# PACE: Post-Causal Entropy Modeling for Learned LiDAR Point Cloud Compression

**Jiahao Zhu** [* 1]  **Kang You** [* 2]  **Dandan Ding** [1]  **Zhan Ma** [2]

## Abstract

LiDAR point cloud compression is vital for autonomous systems to handle massive data from high-resolution sensors. While learned entropy modeling built upon octree structures yields high compression gains, it faces two critical bottlenecks: 1) prohibitive latency, particularly during decoding, caused by causal, multi-stage context modeling; and 2) a rigid performance-latency trade-off, preventing a single model from adapting to varying constraints. These limitations stem from the tight coupling between the context aggregation backbone and probability prediction. To address this, we propose PACE, a new framework that reformulates ancestral context aggregation as a non-causal backbone and confines causality to a lightweight, stage-scalable predictor, eliminating repetitive backbone executions and reducing computational overhead. The predictor supports an arbitrary number of prediction stages, enabling seamless adaptation across diverse performance-latency trade-offs without reloading parameters. Experiments demonstrate that PACE sets a new state-of-the-art in compression efficiency, achieving notable BD-BR savings and reducing decoding latency by over 90% in autoregressive mode, making it attractive for practical applications.

## 1. Introduction

Light detection and ranging (LiDAR) sensors are widely deployed in autonomous driving and robotics to capture continuous high-fidelity 3D representations (Wang et al., 2022; Abbasi et al., 2023; Chen et al., 2024). Outperforming passive vision sensors in range, accuracy, and illumination robustness, LiDAR is essential for localization, mapping, and obstacle detection (Lu et al., 2019; Debeunne & Vivet, 2020; Li et al., 2025). Yet, the high resolution required for autonomy generates enormous data volumes, creating bottlenecks in memory, bandwidth, and computation. Consequently, efficient LiDAR point cloud compression (LPCC) is indispensable to facilitate storage, transmission, and collaborative perception (Wang et al., 2025b; Jing et al., 2026).

Octree-based methods (Zhang et al., 2024) are a cornerstone of the LPCC task, as their hierarchical partitioning effectively translates the problem of 3D geometry compression into the compression of a tree structure. Specifically, an octree recursively dissects the 3D volume into eight subspaces. In this structure, the geometric information of any given node is abstracted into an 8-bit occupancy code, a bitmap indicating the presence of points within its eight children. Consequently, lossless point cloud compression is equivalent to accurately modeling and entropy-coding these occupancy sequences.

Building upon this, modern learning-based approaches (Fu et al., 2022; Song et al., 2023; Jin et al., 2024; Wang et al., 2025d;c) employ deep neural networks to model level-wise occupancy dependencies, utilizing previously encoded levels as priors. Despite this shared reliance on inter-level information, they diverge in intra-level processing: i) one-stage approaches (Cui et al., 2023; You et al., 2025) ignore sibling dependencies and predict all nodes in the current level in one shot, achieving high parallelism but with degraded coding efficiency; ii) multi-stage frameworks (Fu et al., 2022; Wang et al., 2025c) integrate both ancestor and available sibling nodes to achieve superior compression gains, at the expense of increased computational latency.

Despite their respective advantages, both solutions exhibit inherent limitations in latency-efficiency trade-offs and adaptability, which hinder practical deployment:

- *Linear latency scaling.* Most existing multi-stage solutions adopt a fully-causal framework (see Fig. 1a), where a heavyweight backbone is repeatedly invoked for sequential processing, causing inference latency to grow linearly with the number of stages.

---

[*]Equal contribution [1]School of Information Science and Technology, Hangzhou Normal University, Hangzhou, China. [2]School of Electronic Science and Engineering, Nanjing University, Nanjing, China. Correspondence to: Dandan Ding <DandanDing@hznu.edu.cn>.

*Proceedings of the $43^{rd}$ International Conference on Machine Learning*, Seoul, South Korea. PMLR 306, 2026. Copyright 2026 by the author(s).

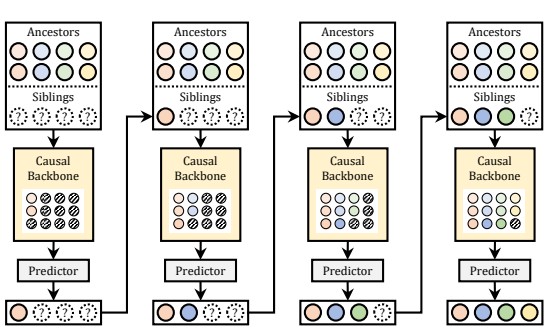 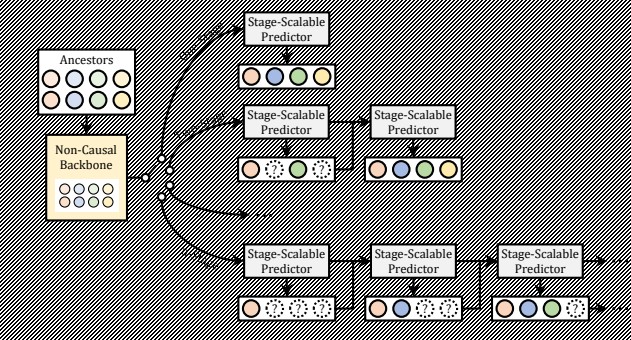

*(a)* Existing *fully-causal* modeling                     *(b)* Proposed *post-causal* PACE

*Figure 1.* Octree-based LPCC paradigm. (a) Conventional fully-causal modeling, which demands repeated backbone and predictor executions due to intra-level causality constraints. (b) Proposed post-causal pipeline, which constrains causality to a lightweight and stage-scalable predictor only, reducing computational overhead and supporting dynamic stage transitions.

- *Inflexible performance-latency trade-off.* Current solutions adopt a fixed-stage design (e.g., one-stage or $n$-stage) and cannot adapt to varying resource budgets, while real applications require dynamic trade-offs between compression efficiency (e.g., archiving) and latency (e.g., interactive streaming).

To address these challenges, we propose PACE, an octree-based LPCC framework featuring a *post-causal* and *stage-scalable* entropy modeling paradigm, as illustrated in Fig. 1b. Specifically, we decouple the inter-level (i.e., ancestors) feature extraction into a non-causal backbone and impose causality only on the intra-level (i.e., siblings) prediction. By shifting causal constraints from the heavyweight backbone to the lightweight predictor, PACE eliminates repetitive backbone executions inherent in conventional fully-causal modeling, substantially reducing computational overhead. Furthermore, we devise a stage-scalable predictor with a *"one-size-fits-all"* capability via elastic causal embedding, which supports an arbitrary number of prediction stages with seamless transitions. This enables on-the-fly reconfiguration in applications for varying performance-latency trade-offs without altering model parameters.

Extensive experiments demonstrate that PACE is remarkably effective across various stage configurations. In particular, PACE establishes new state-of-the-art across four representative datasets, including SemanticKITTI (Behley et al., 2019), Ford (Pandey et al., 2011), nuScenes (Caesar et al., 2020), and QNX (Flynn & Lasserre, 2018), achieving substantial BD-BR savings (39.01%~50.82% for one-stage and 45.14%~55.18% for multi-stage) compared to G-PCC. Notably, in autoregressive mode, PACE achieves >20% compression gains and >90% decoding latency reduction compared to a recent learning-based TopNet (Wang et al., 2025d), highlighting the effectiveness of our post-causal design. Overall, PACE offers an efficient underlying framework for LPCC, which is attractive for real applications.

The main contributions of this paper are as follows:

- We propose PACE, a post-causal entropy modeling framework that effectively decouples non-causal inter-level context aggregation from causal intra-level prediction, substantially reducing decoding overhead.

- We introduce a stage-scalable predictor with elastic causal embedding, which supports an arbitrary number of prediction stages within a single model and on-the-fly reconfiguration under dynamic constraints.

- Extensive experiments on diverse benchmarks demonstrate that PACE achieves state-of-the-art compression efficiency and significantly reduces decoding latency, validating its effectiveness and practical value.

## 2. Related Work

Learning-based LiDAR geometry compression mainly follows two routes: sparse tensor-based and octree-based methods, as introduced below.

### 2.1. Sparse Tensor-based Model

Sparse tensor represents a point cloud by partitioning 3D space into unordered voxels while skipping empty ones. By applying 3D sparse convolutions on multiscale sparse representations, methods like SparsePCGC (Wang et al., 2023), Unicorn (Wang et al., 2025a), and UniPCGC (Wang & Gao, 2025) exploit ancestry and sibling correlations, significantly outperforming the traditional G-PCC standard (WG 07 MPEG 3D Graphics Coding and Haptics Coding, 2023). However, these methods often require large kernels to handle voxel sparsity inherent in LiDAR data, increasing computational overhead. Although lightweight variants (You et al., 2025; Yu et al., 2025) accelerate inference, the local receptive fields of 3D convolutions (e.g., $3^3$ kernel size) limit their ability to capture long-range dependencies.

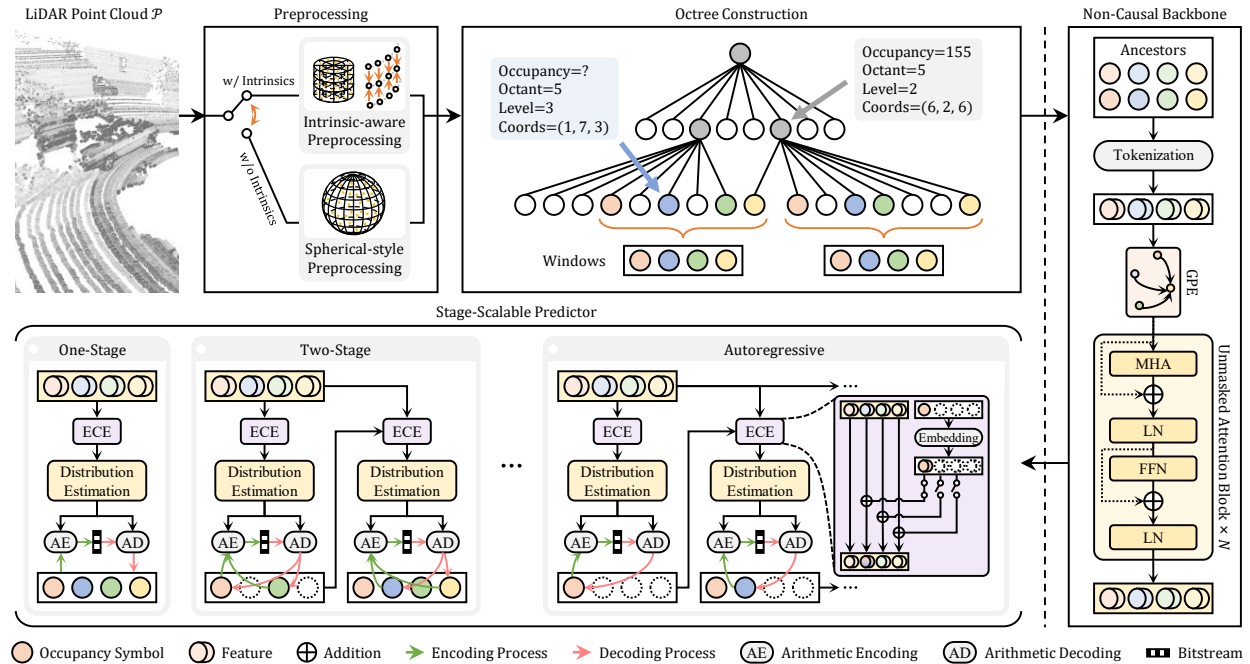

*Figure 2.* Framework of our proposed PACE. The LiDAR point cloud is first preprocessed for efficient octree construction, optionally using intrinsic-aware or spherical-style preprocessing depending on whether sensor intrinsics are available. Then, nodes are organized into windows for our post-causal processing: the non-causal backbone applies inter-level context aggregation, and the stage-scalable predictor exploits intra-level causality, generating the occupancy probabilities for subsequent arithmetic coding. GPE and ECE refer to the introduced graph-based positional encoding and elastic causal embedding, respectively.

## 2.2. Octree-based Model

Due to the inherent sparsity of LiDAR scans, a serialized approach is more conducive to modeling contextual information, as demonstrated in perception tasks (Wu et al., 2024; Jin et al., 2025). Thus, octree-based representations have emerged as a popular paradigm for learned LPCC.

**One-stage approach.** Previous efforts such as Oct-Squeeze (Huang et al., 2020) and OctFormer (Cui et al., 2023) focus on ancestor-to-child dependencies. By discarding sibling contexts, they implement high-speed parallel decoding. However, the absence of intra-level dependencies results in suboptimal compression gains.

**Autoregressive approach.** To improve the coding performance, OctAttention (Fu et al., 2022) introduces sibling dependencies via masked attention, predicting node status on a node-by-node basis. This paradigm is further refined by TopNet (Wang et al., 2025d) and ASRL (Wang et al., 2025c), which utilize convolution-enhanced or sparsity-aware attention to capture long-range dependencies. Despite superior compression performance, their strict serial dependency forces re-executing the heavy backbone for each node, causing prohibitive decoding latency.

**Multi-stage approach.** To balance compression efficiency and computational cost, EHEM (Song et al., 2023) and ECM-OPCC (Jin et al., 2024) devise multi-stage context

modeling, partitioning nodes into disjoint sets and decoding them stage by stage. GAEM (Cui et al., 2025) further utilizes graph-driven attention in context aggregation to capture structural correlations. Despite improving R-D performance, they are constrained by a rigid coupling between context aggregation backbone and prediction head: the backbone must be re-executed for each stage, and the performance-latency trade-off is fixed once the model is trained.

**In summary,** one-stage models prioritize low latency but sacrifice compression gains, whereas multi-stage and autoregressive models achieve higher gains at the cost of linearly increased decoding latency. A robust paradigm that overcomes this limitation while offering flexible performance-latency trade-offs remains an open challenge.

## 3. Methodology

The framework of PACE is illustrated in Fig. 2, consisting of three stages: octree construction using intrinsic-aware or spherical-style preprocessing, non-causal backbone for inter-level context aggregation, and stage-scalable predictor to predict the occupancy distribution for entropy coding.

### 3.1. Intrinsic-aware Preprocessing

Let $\mathcal{P}=\{(x_i, y_i, z_i)\}_{i=1}^N$ be the input LiDAR point cloud, where each point $(x_i, y_i, z_i)$ represents the Cartesian co-

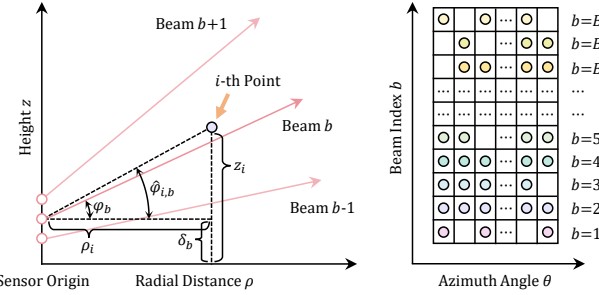

*Figure 3.* Beam index mapping. Left: The $i$-th point is mapped to a discrete beam by minimizing the angular residual between its theoretical pitch $\hat{\varphi}_{i,b}$ and calibrated pitch $\varphi_b$. Right: The structured hybrid cylindrical-beam representation, offering a more hardware-aligned alternative to standard cylindrical projections.

ordinates. Unlike prior methods that directly project Cartesian coordinates $(x_i, y_i, z_i)$ into a cylindrical system $(\rho_i, \theta_i, z_i)$ (Gao et al., 2023), we further utilize sensor intrinsics (e.g., laser pitch angles and vertical offsets) to map the vertical component $z_i$ to a discrete beam index $b_i$, yielding a hybrid cylindrical-beam representation $(\rho_i, \theta_i, b_i)$. The discretization of $b_i$ inherently reduces representation complexity, and can be reversed via sensor calibration during decoding to recover the original $z_i$ coordinates.

**Cylindrical coordinate conversion.** We first convert the input point cloud from Cartesian coordinates $(x_i, y_i, z_i)$ to cylindrical coordinates $(\rho_i, \theta_i, z_i)$ via:

$$\rho_i = \sqrt{x_i^2 + y_i^2}, \ \theta_i = \arctan\left(\frac{y_i}{x_i}\right), \ \text{and} \ z_i = z_i, \quad (1)$$

where $\rho_i$ and $\theta_i$ refer to the radial distance (or planar distance) and the azimuth angle of the $i$-th point, respectively.

**Beam index mapping.** Define the sensor intrinsics by the laser pitch angles $\Phi = \{\varphi_b\}_{b=1}^B$ and per-beam vertical offsets $\Delta = \{\delta_b\}_{b=1}^B$, where $B$ denotes the total number of laser beams (e.g., 32 or 64) and $b$ is the beam index. As depicted in Fig. 3, for each point, we calculate the theoretical pitch angle corresponding to the $b$-th beam, mathematically:

$$\hat{\varphi}_{i,b} = \arctan\left(\frac{z_i - \delta_b}{\rho_i}\right), \ \forall b \in \{1, \ldots, B\}. \quad (2)$$

The point is then assigned to the beam index $b_i$ that minimizes the angular residual, i.e.,

$$b_i = \operatorname*{argmin}_{b \in \{1, \ldots, B\}} \|\hat{\varphi}_{i,b} - \varphi_b\|_2^2. \quad (3)$$

This yields the cylindrical-beam representation $(\rho_i, \theta_i, b_i)$, which is used in PACE to construct an octree structure. However, in scenarios where accurate intrinsics are unavailable (e.g., the LiDAR scans in SemanticKITTI (Behley et al., 2019)), we adopt a standard multi-level spherical processing following the previous work (Luo et al., 2024), thereby ensuring applicability across diverse datasets.

### 3.2. Octree-based Deep Entropy Model

**Octree representation.** Building upon the intrinsic-aware coordinate transformation, the point cloud is discretized into a hierarchical multi-level octree structure. Let $L$ denote the maximum octree depth. For any given level $l \in \{1, \ldots, L\}$, the occupancy sequence $\mathcal{O}^l$ is derived by collecting all non-empty nodes via breadth-first traversal:

$$\mathcal{O}^l = \{o_1^l, o_2^l, \ldots, o_{N_l}^l\}, \quad (4)$$

where $o_i^l \in \{1, \ldots, 255\}$ represents the occupancy symbol of the $i$-th node at level $l$, and $N_l$ denotes the total number of nodes at this level. Consequently, the complete geometry of the LiDAR scan is encapsulated by the concatenated level-wise occupancy sequences, denoted as $\mathcal{O} = \{\mathcal{O}^1, \ldots, \mathcal{O}^L\}$.

**Probabilistic entropy modeling.** To compress the occupancy sequence $\mathcal{O}$, a deep parametric model is constructed to approximate the probability distribution $p(\mathcal{O})$. Theoretically, minimizing the compression bitrate $\mathcal{R}$ is equivalent to minimizing the cross-entropy between the estimated distribution $p(\mathcal{O})$ and the actual distribution $q(\mathcal{O})$:

$$\mathcal{R} = \mathbb{E}_{\mathcal{O} \sim q(\mathcal{O})}[-\log_2 p(\mathcal{O})]. \quad (5)$$

To effectively model the complex dependencies within the octree, the probability $p(\mathcal{O})$ is factorized into a coarse-to-fine chain, which incorporates two dependency paradigms:

*i) Inter-level prediction* utilizes ancestral information from coarser levels $(\mathcal{O}^{<l})$ to progressively predict the occupancy distribution of the current level $l$:

$$p(\mathcal{O}) = \prod_{l=1}^L p\left(\mathcal{O}^l \mid \mathcal{O}^{<l}\right). \quad (6)$$

*ii) Intra-level prediction* decomposes the current level into $S$ stages, denoted as $\mathcal{O}^l = \{\mathcal{O}_1^l, \ldots, \mathcal{O}_S^l\}$, and predicts the current stage conditioned on the preceding ones:

$$p\left(\mathcal{O}^l | \mathcal{O}^{<l}\right) = p\left(\mathcal{O}_1^l, \mathcal{O}_2^l, \ldots, \mathcal{O}_S^l | \mathcal{O}^{<l}\right), \quad (7)$$
$$= p\left(\mathcal{O}_1^l | \mathcal{O}^{<l}\right) \prod_{s=2}^S p\left(\mathcal{O}_s^l | \mathcal{O}_{<s}^l, \mathcal{O}^{<l}\right).$$

**Remark.** In deep entropy models, inter-level dependencies are strictly dictated by the octree's hierarchical structure. Therefore, this work concentrates on intra-level stage-wise causality (i.e., the sequential dependencies between different stages) in multi-stage (including autoregressive) processing.

### 3.3. Non-Causal Backbone

Unlike prior works that fuse inter-level $(\mathcal{O}^{<l})$ and intra-level $(\mathcal{O}_{<s}^l)$ contexts before the causal backbone (either via raw occupancy symbols (Fu et al., 2022; Wang et al., 2025d) or feature representations (Song et al., 2023; Luo et al., 2024)), we propose to aggregate inter-level context using a non-causal backbone and introduce interaction between inter- and intra-level contexts within a lightweight predictor.

**Window partition.** To facilitate efficient attention-based feature interaction, the occupancy sequence $\mathcal{O}^l$ at level $l$ is partitioned into non-overlapping windows of size $W$. Specifically, the $m$-th window $\mathcal{O}^{l,m}$ is defined as:

$$\mathcal{O}^{l,m} = \left\{ o_i^l \mid (m-1)W < i \leq mW \right\}. \quad (8)$$

The superscripts $l$ and $m$ are omitted in the following for brevity since subsequent operations are performed independently within each window.

**Context tokenization.** For $i$-th octree node in the window, we first project its inter-level context into a vectorized token $\mathbf{e}_i$, which fuses the octant $t_i$, the level $l$, and the embeddings of $G$ generations of ancestors $\{o_i^{(g)}\}_{g=1}^G$:

$$\mathbf{e}_i = \left( \oplus_{g=1}^G \text{Emb}(o_i^{(g)}) \right) \oplus \text{Emb}(t_i) \oplus \text{Emb}(l), \quad (9)$$

where $\text{Emb}(\cdot)$ denotes the embedding layer (implemented as a learnable lookup table); $\text{MLP}(\cdot)$ means multi-layer perceptron; $\oplus$ denotes concatenation. We empirically set $G$ to 3 in our experiments.

**Graph-based positional encoding.** We introduce graph-based positional encoding (GPE) to explicitly incorporate 3D structural priors in our backbone. Specifically, for each node $i$, let $\mathbf{c}_i \in [-1, 1]$ denote its normalized coordinates[1], we first fuse the context token $\mathbf{e}_i$ with the coordinates $\mathbf{c}_i$:

$$\mathbf{e}_i' = \text{MLP}(\mathbf{e}_i) + \text{MLP}(\mathbf{c}_i). \quad (10)$$

To capture local dependencies, we construct a $k$-NN graph $\mathcal{N}(i)$ and compute edge features $\mathbf{e}_j'$ via an EdgeConv (Wang et al., 2019) variant, encoding the interaction between node $i$ and its spatial neighbors $j \in \mathcal{N}(i)$:

$$\mathbf{e}_j'' = \text{MLP}\left( \mathbf{e}_i' \oplus \left( \mathbf{e}_j' - \mathbf{e}_i' \right) \right). \quad (11)$$

Finally, neighborhood information is aggregated through a gated mechanism:

$$\bar{\mathbf{e}}_i = \underset{j \in \mathcal{N}(i)}{\text{MAX}} \left\{ \text{MLP}\left( \mathbf{e}_j'' \odot \text{Gate}(\mathbf{e}_j'') \right) \right\}, \quad (12)$$

where $\text{Gate}(\cdot)$ denotes a Linear layer with SiLU activation; $\odot$ means the Hadamard product; MAX performs channel-wise max-pooling over the local neighborhood $j \in \mathcal{N}(i)$.

**Unmasked Attention.** Following GPE, we apply unmasked scaled dot-product attention within each window, as shown on the right of Fig. 2. Specifically, let the input feature be $\mathcal{A}^{(0)} = \{\bar{\mathbf{e}}_i\}_{i=1}^W$, then the updated feature $\mathcal{A}^{(i+1)}$ is computed by querying all nodes within the window:

$$\tilde{\mathcal{A}}^{(i)} = \text{LayerNorm}\left( \mathcal{A}^{(i)} + \text{MHA}(\mathcal{A}^{(i)}) \right), \quad (13)$$

$$\mathcal{A}^{(i+1)} = \text{LayerNorm}\left( \tilde{\mathcal{A}}^{(i)} + \text{FFN}(\tilde{\mathcal{A}}^{(i)}) \right), \quad (14)$$

---

[1]Depending on the availability of intrinsics, the coordinates are represented either as $(\rho_i, \theta_i, b_i)$ or as spherical coordinates.

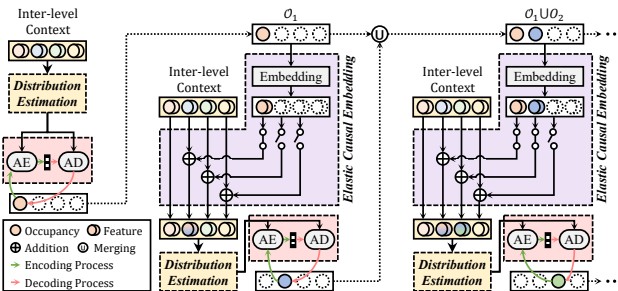

*Figure 4.* Stage-scalable predictor. The inter-level context yielded by the non-causal backbone is shared across all stages, and the intra-level context ($O_{<s}, s \in [1, S]$) from previous stages is embedded and fused to predict the target stage $O_s$. The autoregressive mode is presented as an illustrative example.

where $\text{MHA}(\cdot)$ and $\text{FFN}(\cdot)$ denote multi-head attention and a feed-forward network, respectively. By stacking $N$ such layers, the final refined feature serves as the non-causal inter-level context for the subsequent stage-scalable predictor.

### 3.4. Stage-Scalable Predictor

Building upon the non-causal backbone that extracts inter-level context, we introduce a stage-scalable predictor with flexible stage-wise causality, achieving dynamic trade-offs between compression efficiency and computational latency.

**Stage-wise decomposition.** Given a window of occupancy symbols $\mathcal{O} = \{o_i\}_{i=1}^W$ and corresponding inter-level context $\mathcal{A} = \{\mathbf{a}_i\}_{i=1}^W$, we uniformly partition $\mathcal{O}$ into $S$ stages via:

$$\mathcal{O}_s = \{o_{k \times S + s} \mid k \times S + s \leq W, k \in \mathbb{N}_0\}. \quad (15)$$

By varying $S$, the predictor can span a spectrum of coding paradigms. E.g., when $S=1$, the entire window is treated as a single stage $\mathcal{O}_1 = \{o_i\}_{i=1}^W$ for fully parallel processing; when $S=W$, each stage contains exactly one node, i.e., $\mathcal{O}_s = \{o_s\}$, resulting in a fully autoregressive mode. We transmit $S$ to the decoder for consistent stage decomposition.

**Elastic causal embedding.** As shown in Fig. 4, for each node $i$ in the $s$-th stage, we fuse the inter-level context $\mathbf{a}_i$ with the preceding context from previous stages $\mathcal{O}_{<s}$:

$$\mathbf{f}_i^{(s)} = \mathbf{a}_i + \mathbb{I}(o_{i-1} \in \bigcup \mathcal{O}_{<s}) \cdot \text{Emb}(o_{i-1}), \quad (16)$$

where the indicator function $\mathbb{I}(\cdot)$ ensures that sibling $o_{i-1}$ is only utilized if it was decoded in a preceding stage. Note that when $s=1$, $\mathbf{f}_i^{(s)}$ is equivalent to $\mathbf{a}_i$, as the set of preceding stages $\mathcal{O}_{<s}$ is empty.

**Distribution estimation.** To capture local dependencies within the fused features $\mathcal{F}^{(s)} = \{\mathbf{f}_i^{(s)}\}_{i=1}^W$, we employ a Selective State Space Model (SSM) instantiated by a Mamba (Gu & Dao, 2023) block. The SSM facilitates efficient context aggregation across the sequence:

$$\tilde{\mathcal{F}}^{(s)} = \text{SSM}(\mathcal{F}^{(s)}). \quad (17)$$

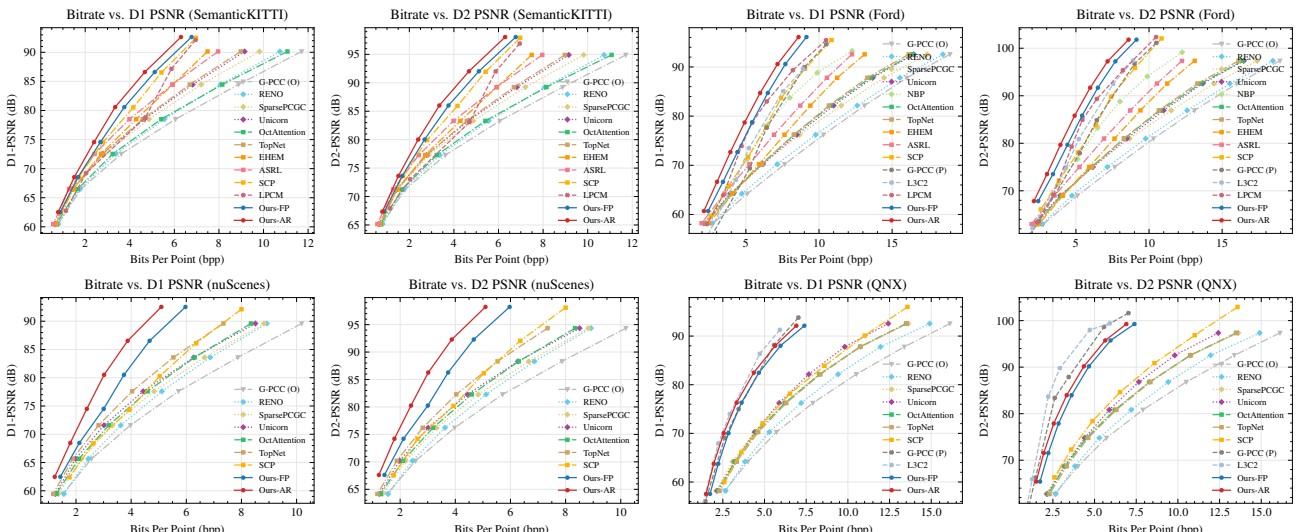

*Figure 5.* Comparison of rate-distortion (R-D) performance across SemanticKITTI, Ford, nuScenes, and QNX datasets. G-PCC (O) and G-PCC (P) represent the Octree and Predgeom configurations of G-PCC, respectively.

The occupancy distribution for each symbol $o_i \in \mathcal{O}_s$ is then estimated via $\tilde{\mathbf{f}}_i^{(s)} \in \tilde{\mathcal{F}}^{(s)}$:

$$p(o_i) = \text{Softmax}\left(\text{MLP}\left(\tilde{\mathbf{f}}_i^{(s)}\right)\right), \qquad (18)$$

where $p(o_i)$ serves as the categorical distribution for the arithmetic coder to compress $o_i$.

## 4. Experiments and Analysis

### 4.1. Experimental Settings

**Datasets.** We evaluate the performance of PACE on four representative datasets: SemanticKITTI (Behley et al., 2019), Ford (Pandey et al., 2011), nuScenes (Caesar et al., 2020), and QNX (Flynn & Lasserre, 2018).

- **SemanticKITTI** is acquired using a Velodyne HDL-64E LiDAR sensor and comprises 22 point cloud sequences with a total of 43,504 frames. Each frame contains about 120k points. Following prior works (Song et al., 2023; Wang et al., 2025d), sequences #00∼#10 are used for training and #11∼#21 for testing.

- **Ford** is also collected using a Velodyne HDL-64E LiDAR sensor. Under the common test condition (CTC) defined by MPEG G-PCC (WG 07 MPEG 3D Graphics Coding and Haptics Coding, 2024), three Ford sequences are used, each having 1,500 frames at 1 mm precision. The first sequence is used for training and the other two for testing.

- **nuScenes** is a large-scale autonomous driving dataset collected via a Velodyne HDL-32E LiDAR sensor. We select 6,000 frames for training and 450 for testing.

- **QNX** is a LiDAR dataset recommended by the MPEG G-PCC CTC, collected using a Velodyne VLP16 sensor at 1 mm precision. It contains four sequences with approximately 30k points per frame. Two sequences are used for training and the remaining two for testing.

All samples are voxelized following prior works (Song et al., 2023; Luo et al., 2024). Specifically, we set the quantization depth to $L$=16 for raw SemanticKITTI and nuScenes, with scaling factors of $\frac{400}{2^L-1}$ and $\frac{450}{2^L-1}$, respectively. Ford and QNX are already voxelized at 1 mm precision. We vary $L$ to evaluate the model across different bitrates.

**Baseline Methods.** We compare PACE against existing LPCC methods: rules-based MPEG G-PCC (WG 07 MPEG 3D Graphics Coding and Haptics Coding, 2023) TMC13v23 (Octree and Predgeom) and TMLv4 (a.k.a. L3C2 (Sébastien & Jonathan, 2021)); multiscale sparse tensor-based methods RENO (You et al., 2025), SparsePCGC (Wang et al., 2023), Unicorn (Wang et al., 2025a), and NBP (Liu et al., 2026); octree-based methods OctAttention (Fu et al., 2022), TopNet (Wang et al., 2025d), EHEM (Song et al., 2023), ASRL (Wang et al., 2025c), and SCP (Luo et al., 2024); predtree-based method LPCM (Sun et al., 2025). All methods are evaluated under identical conditions.

**Evaluation Metrics.** We use point-to-point (D1) and point-to-plane (D2) PSNR to measure distortion and bits per point (bpp) to measure bitrate. PSNR peak values are set to 59.70 for SemanticKITTI and nuScenes and 30,000 for Ford and QNX. The Bjøntegaard Delta Bitrate (BD-BR) (Bjøntegaard, 2001) is used to assess rate-distortion (R-D) performance, which quantifies the average bitrate change between two codecs at equivalent objective quality, where lower is better.

*Table 1.* Compression performance comparison. BD-BR (%) is reported against the G-PCC (Octree) anchor. Ours-FP and Ours-AR denote our fully parallel (one-stage) and autoregressive modes, respectively. The top three results are highlighted as first, second and third.

| METHOD | PUB. | KITTI | | FORD | | NUSCENES | | QNX | |
|---|---|---|---|---|---|---|---|---|---|
| | | D1 ↓ | D2 ↓ | D1 ↓ | D2 ↓ | D1 ↓ | D2 ↓ | D1 ↓ | D2 ↓ |
| *Sparse tensor-based models* | | | | | | | | | |
| RENO | *CVPR 2025* | -06.21% | -06.20% | -05.02% | -05.02% | -08.53% | -08.58% | -07.62% | -07.64% |
| SPARSEPCGC | *TPAMI 2023* | -18.64% | -18.63% | -16.08% | -16.08% | -14.84% | -14.86% | -21.40% | -21.40% |
| UNICORN | *TPAMI 2025* | -22.26% | -22.26% | -18.51% | -18.50% | -21.55% | -21.56% | -23.84% | -23.86% |
| NBP | *TPAMI 2026* | N/A | N/A | -39.71% | -41.19% | N/A | N/A | N/A | N/A |
| *Octree-based models* | | | | | | | | | |
| OCTATTENTION | *AAAI 2022* | -08.93% | -08.93% | -18.97% | -18.97% | -18.90% | -18.92% | -19.90% | -19.91% |
| TOPNET | *CVPR 2025* | -22.58% | -22.57% | -18.77% | -18.77% | -28.20% | -28.22% | -18.59% | -18.60% |
| EHEM | *CVPR 2023* | -26.44% | -26.41% | -24.60% | -24.60% | N/A | N/A | N/A | N/A |
| ASRL | *SPL 2025* | -31.98% | -31.97% | -32.43% | -32.43% | N/A | N/A | N/A | N/A |
| SCP | *AAAI 2024* | -35.72% | -37.87% | -40.26% | -44.45% | -20.67% | -25.42% | -22.66% | -29.58% |
| *Predtree-based models* | | | | | | | | | |
| G-PCC (PREDGEOM) | *MPEG 2023* | N/A | N/A | -45.24% | -48.53% | N/A | N/A | -53.21% | -66.15% |
| L3C2 | *MPEG 2025* | N/A | N/A | -45.67% | -50.06% | N/A | N/A | -57.76% | -71.57% |
| LPCM | *ArXiv 2025* | -28.91% | -25.94% | -48.30% | -50.07% | N/A | N/A | N/A | N/A |
| OURS-FP | *This Paper* | -39.01% | -41.00% | -50.19% | -55.02% | -39.77% | -42.91% | -50.82% | -57.86% |
| OURS-AR | *This Paper* | **-45.14%** | **-46.97%** | **-55.07%** | **-59.51%** | **-50.38%** | **-52.95%** | -55.18% | -61.73% |

**Implementation Details.** We train our models using the AdamW optimizer (Loshchilov & Hutter, 2019) with an initial learning rate of $5 \times 10^{-4}$. The training duration varies by dataset: 10 epochs for SemanticKITTI and QNX, 25 epochs for nuScenes, and 50 epochs for Ford. All experiments are conducted on a workstation equipped with an NVIDIA RTX 4090 GPU and an Intel i9-13900K CPU.

## 4.2. Performance Evaluation

Figure 5 and Table 1 present the BD-BR performance of various methods across four datasets. Under both one-stage mode (i.e., fully parallel mode, termed Ours-FP) and the autoregressive mode (Ours-AR), PACE consistently ranks as the top or among the top performer across all datasets and distortion metrics, demonstrating a clear advantage over both traditional rules-based and learning-based approaches. While recent octree-based models (e.g., TopNet (Wang et al., 2025d), EHEM (Song et al., 2023), and SCP (Luo et al., 2024)) already surpass the G-PCC (Octree) baseline, PACE further improves BD-BR by a substantial margin.

Note that rules-based predtree models are restricted to scans with LiDAR intrinsics, as prediction tree construction relies heavily on these parameters. Thus, they apply only to intrinsics-available datasets such as Ford and QNX, often outperforming octree-based models. However, by incorporating intrinsic-aware preprocessing, PACE equips the octree with the ability to leverage intrinsic data, leading to significant compression improvements in these scenarios. For example, PACE attains comparable gains with G-PCC (Predgeom) and L3C2 on QNX and surpasses them on Ford. PACE also outperforms LPCM, a learned predtree method.

*Table 2.* Complexity comparison. We compare our method against open-source octree-based baselines in terms of model size, memory usage, and runtime (encoding/decoding) at octree level $L$=14.

| METHOD | PARAM. ↓ | MEM. ↓ | ENC. ↓ | DEC. ↓ |
|---|---|---|---|---|
| OCTATTENTION | 4.23M | 0.31GB | 2.40s | 282.04s |
| TOPNET | 3.37M | 0.30GB | 2.46s | 290.26s |
| SCP | 24.51M | 1.08GB | 8.94s | 9.01s |
| OURS-FP | 10.57M | 0.58GB | 1.72s | 1.55s |
| OURS-AR | 10.57M | 0.58GB | 10.89s | 16.36s |

## 4.3. Complexity Comparison

PACE complements its R-D gains with affordable computational complexity. As presented in Table 2, the one-stage mode (Ours-FP) achieves an optimal speed-performance trade-off, delivering higher compression accuracy while outperforming competing methods in both encoding and decoding speeds. Furthermore, the autoregressive mode (Ours-AR) reduces decoding latency by over 90% compared to existing autoregressive models such as OctAttention and TopNet, a leap attributed to our post-causal design.

Currently, our implementation remains a prototype with encoding-time overhead in AR mode as the main bottleneck (10.89 s). In principle, as all information is available during encoding, the stage-scalable predictor can be parallelized to further improve speed (theoretically comparable to the FP mode), which we leave for our future work.

## 4.4. Ablation Study

Ablation studies evaluate the contribution of each component using the Ford dataset recommended by MPEG CTC.

**Preprocessing.** LiDAR's mechanical scanning often requires preprocessing (e.g., coordinate transformation) to optimize octree representation for higher compression gains. For instance, SCP (Luo et al., 2024) uses a spherical-coordinate-based octree. We evaluate PACE across various preprocessing methods in Fig. 6. As observed, the SCP scheme (labeled "Sph.") surpasses the direct octree construction in the Cartesian coordinate system (labeled "Cart."). Further, our intrinsic-aware approach achieves the best performance by better leveraging sensor intrinsics.

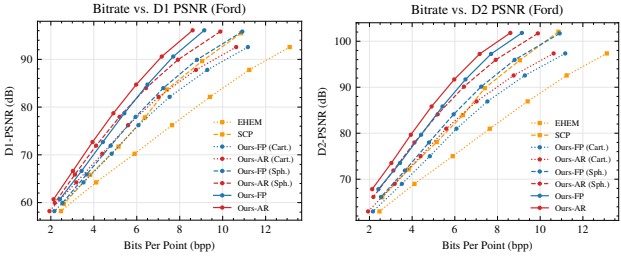

*Figure 6.* Ablation on preprocessing methods. "Sph." and "Cart." denote spherical- and Cartesian-based octrees, respectively.

**Fully-causal vs. Post-causal modeling.** The strategy of concatenating ancestors and siblings before a masked backbone (termed fully-causal modeling) remains a common practice in existing works (Wang et al., 2025d;c). We implement this strategy in PACE for comparison. Figure 7 illustrates that the fully-causal PACE suffers from high decoding latency due to repeated backbone computations (e.g., ~90× slower at level $L$=14), and its masking scheme causes information loss by occluding ancestral features, resulting in inferior performance compared to the post-causal version (-52.34% vs. -55.07%). In addition, the post-causal pipeline allows on-the-fly reconfiguration of prediction stages. For instance, a single trained PACE model can be directly applied to any number of stages without parameter reloading. Conversely, the fully-causal model must be specifically retrained for each new configuration, greatly increasing storage and computational overhead.

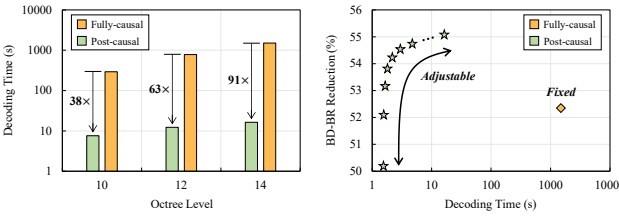

*Figure 7.* Evaluation of fully-causal and post-causal modeling. Left: Decoding latency comparison in the autoregressive (AR) mode of PACE. Right: Efficiency-complexity Pareto front, where complexity is measured as decoding time at octree level 14.

**Graph-based positional encoding (GPE).** Table 3 evaluates the impact of GPE. Notably, GPE adds negligible

overhead while consistently improving compression: BD-BR reduction is improved from -38.22% to -50.19% in our FP mode and from -46.14% to -55.07% in our AR mode. These results suggest that modeling 3D geometric relations via graphs effectively complements the serialized octree order, yielding more informative context.

*Table 3.* Ablation study on graph-based positional encoding

| CONFIG | OURS-FP | | OURS-AR | |
|---|---|---|---|---|
| | PARAM. ↓ | BD-BR ↓ | PARAM. ↓ | BD-BR ↓ |
| W/O GPE | 10.24M | -38.22% | 10.24M | -46.14% |
| W/ GPE | 10.57M | -50.19% | 10.57M | -55.07% |

**Stage-scalable prediction.** To evaluate the flexibility and efficiency of our stage-scalable predictor, we analyze the performance-latency trade-offs by varying the number of stages $S$ during inference. As illustrated in Table 4, increasing $S$ from 1 to 16 consistently enhances BD-BR performance (from -50.19% to -54.23%) by capturing denser intra-level dependencies. Notably, as our post-causal design executes the heavy backbone only once, the computational overhead grows only marginally, with decoding time increasing from 1.55 s to 2.18 s from 1-stage to 16-stage. Although the fully autoregressive variant achieves the best BD-BR (-55.07%), it suffers from longer decoding latency due to the node-by-node prediction; therefore, it mainly serves as an upper-bound reference. In practical applications, the number of stages in PACE can be dynamically configured for balanced trade-offs.

*Table 4.* On-the-fly reconfiguration of prediction stages in PACE

| CONFIG | MEM. ↓ | ENC. ↓ | DEC. ↓ | BD-BR ↓ |
|---|---|---|---|---|
| 1-STAGE | 0.58GB | **1.72s** | **1.55s** | -50.19% |
| 2-STAGE | 0.58GB | 1.76s | 1.57s | -52.09% |
| 4-STAGE | 0.58GB | 1.81s | 1.66s | -53.16% |
| 8-STAGE | 0.58GB | 1.94s | 1.80s | -53.81% |
| 16-STAGE | 0.58GB | 2.20s | 2.18s | -54.23% |
| AUTOREGRESSIVE | 0.58GB | 10.89s | 16.36s | **-55.07%** |

## 5. Conclusion

This paper introduces PACE, an LPCC framework that addresses the rigid trade-off between compression gain and computational overhead via a post-causal paradigm. By decoupling inter-level context aggregation into a non-causal backbone and restricting causality to a stage-scalable predictor, we eliminate redundant computations and improve compression gains. Under the post-causal paradigm, PACE achieves state-of-the-art efficiency across four benchmarks while reducing decoding latency by over 90% in autoregressive mode. For practical deployment, our PACE offers the flexibility required by real systems and supports seamless adaptation to varying performance-latency constraints.

## Acknowledgments

This research was supported by National Natural Science Foundation of China (Grant Nos. 62431011, 62571174) and Natural Science Foundation of Jiangsu Province (Grant No. BK20243038). The authors gratefully acknowledge the support of the Advanced Computation Center of Hangzhou Normal University and thank the anonymous reviewers for their valuable comments on earlier drafts of this paper.

## Impact Statement

This paper presents work whose goal is to advance the field of Machine Learning. There are many potential societal consequences of our work, none which we feel must be specifically highlighted here.

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

## A. Dataset Details

Four widely accepted datasets in LiDAR-related studies and standardization activities, including SemanticKITTI, Ford, nuScenes, and QNX, are used for training and evaluation in this work. A dataset overview is provided in the main manuscript. Here we detail the exact train/test splits for reproducibility, especially for nuScenes and QNX.

- **nuScenes.** We construct a fixed, scene-disjoint split, and deterministically sample frames within each scene. The nuScenes dataset is organized into ten subsets, each containing 85 scenes. For training, we extract the first 100 frames from the first 12 scenes in the first five subsets, resulting in 6,000 frames. For testing, we select the first 90 frames from the first scene of each of the last five subsets, yielding 450 frames.

- **QNX.** We follow the MPEG G-PCC CTC (WG 07 MPEG 3D Graphics Coding and Haptics Coding, 2024) sequences and adopt a sequence-level split. QNX contains four sequences: *Junction Approach* (74 frames), *Junction Exit* (74 frames), *Motorway Bridge* (811 frames), and *Navigating Turns* (300 frames). We use *Motorway Bridge* and *Navigating Turns* for training, and *Junction Approach* and *Junction Exit* for testing.

We visualize typical samples from these datasets in Fig. 1.

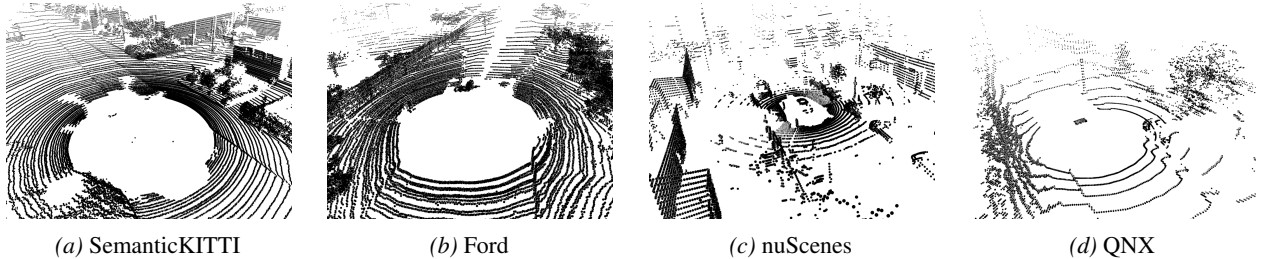

| *(a)* SemanticKITTI | *(b)* Ford | *(c)* nuScenes | *(d)* QNX |

*Figure 1.* Visualization of samples in SemanticKITTI, Ford, nuScenes, and QNX LiDAR point cloud datasets.

## B. More Ablation Studies

### B.1. Design Choices for Stage-Scalable Predictor

The proposed stage-scalable predictor is not tied to a specific architecture. It only requires a causal sequence model to aggregate previously decoded context before probability estimation. We adopt Mamba as the default instantiation because it offers a favorable balance between computational efficiency and context modeling capacity. To justify this design choice, we compare Mamba with a lightweight MLP predictor and an Attention-based predictor in AR mode on the Ford dataset.

Table 1 shows that different context aggregation mechanisms exhibit distinct rate-complexity trade-offs. The MLP predictor incurs the lowest computational overhead, but under our elastic causal embedding it can only exploit limited local context, such as the immediately preceding token. This restricts its ability to model complex conditional distributions for entropy coding and therefore leads to inferior compression performance. The Attention-based predictor captures contextual dependencies more effectively, but its sequential autoregressive execution introduces substantial latency and memory overhead, even with a lightweight single-layer design. In contrast, Mamba achieves the best BD-BR among the compared variants while maintaining the same memory footprint as the MLP predictor and a much lower runtime than Attention. These results support our choice of Mamba as the default stage-scalable predictor, since it preserves strong causal context modeling while remaining practical for efficient LiDAR point cloud compression.

*Table 1.* Comparison of different context aggregation modules for the stage-scalable predictor

| CONFIG | MEM. ↓ | ENC. ↓ | DEC. ↓ | BD-BR ↓ |
|---|---|---|---|---|
| MLPS | 0.58GB | 5.04s | 8.18s | -52.49% |
| ATTENTION | 2.68GB | 33.34s | 40.95s | -54.83% |
| MAMBA | 0.58GB | 10.89s | 16.36s | -55.07% |

## B.2. More Ablation Studies on Stage-Scalable Predictor

In our main manuscript, we conduct an ablation study on our stage-scalable predictor on the Ford (64-beam) dataset. In this supplementary material, we extend the cross-stage evaluation to two additional LiDAR datasets with different beam configurations: nuScenes (32-beam) and QNX (16-beam). The corresponding results are reported in Table 2 and Table 3. It is observed that our PACE exhibits a similar trend on Ford, nuScenes, and QNX. These ablation experiments sufficiently confirm the effectiveness of our stage-scalable predictor across diverse scenarios.

*Table 2.* On-the-fly reconfiguration of prediction stages on nuScenes (32-beam)

| CONFIG | MEM. ↓ | ENC. ↓ | DEC. ↓ | BD-BR ↓ |
|---|---|---|---|---|
| 1-STAGE | 0.53GB | **0.58s** | **0.54s** | -39.77% |
| 2-STAGE | 0.53GB | 0.60s | 0.57s | -44.36% |
| 4-STAGE | 0.53GB | 0.62s | 0.62s | -46.82% |
| 8-STAGE | 0.53GB | 0.68s | 0.69s | -48.18% |
| 16-STAGE | 0.53GB | 0.79s | 0.90s | -48.98% |
| AUTOREGRESSIVE | 0.53GB | 6.61s | 9.84s | **-50.38%** |

*Table 3.* On-the-fly reconfiguration of prediction stages on QNX (16-beam)

| CONFIG | MEM. ↓ | ENC. ↓ | DEC. ↓ | BD-BR ↓ |
|---|---|---|---|---|
| 1-STAGE | 0.52GB | **0.33s** | **0.32s** | -50.82% |
| 2-STAGE | 0.52GB | 0.34s | 0.32s | -52.85% |
| 4-STAGE | 0.52GB | 0.36s | 0.34s | -53.94% |
| 8-STAGE | 0.52GB | 0.39s | 0.37s | -54.55% |
| 16-STAGE | 0.52GB | 0.44s | 0.48s | -54.87% |
| AUTOREGRESSIVE | 0.52GB | 3.27s | 4.91s | **-55.18%** |

## B.3. Weight Sharing for Stage-Scalable Predictor

We use a single predictor in PACE to support multiple coding modes. This design avoids maintaining a dedicated predictor head for each mode. In this section, we examine whether sharing predictor weights leads to any measurable loss in rate-distortion (R-D) performance, while highlighting the parameter savings.

To this end, we compare two training variants under the same data, optimizer, and training schedule; the only difference is whether predictor weights are shared. More specifically,

(i) **Separate predictors**: the backbone is shared, while each supported mode configuration has its own predictor head;

(ii) **Joint predictor**: a single predictor is shared across all modes (the method used in our PACE).

**Computational Complexity.** As summarized in Table 4, separate predictors increase the parameter count linearly with the number of supported configurations. Concretely, if the backbone has $P_b$ parameters and each predictor head has $P_p$, then joint sharing uses $P_b + P_p$, whereas separate heads use $P_b + P_p \times N$, where $N$ is the number of supported configurations. Obviously, the joint predictor largely saves parameters, which is significant for storage-sensitive tasks. Despite the additional capacity, the RD improvement from separate predictors is negligible: the joint predictor matches the separate-head setting closely across both FP and AR (the BD-BR gap remains very small). This indicates that a single conditioned predictor generalizes well across different stages and does not become the compression performance bottleneck when reconfiguring the number of stages at inference time.

**Compression Performance.** In terms of the coding performance, the joint predictor performs comparably to the separate predictors (e.g., -50.19% vs. -50.68% in FP and -55.07% vs. -55.93% in AR), although the separate predictors employ more parameters.

**Discussion.** A key practical advantage of the joint predictor is its cross-mode generalization capability. During training, we use a mixed-stage strategy in which the stage number $S$ is sampled from a small subset (e.g., $S \in \{1, 2, 4, W\}$), and the same predictor is optimized across these configurations. Remarkably, at inference time, it generalizes effectively to much finer decompositions without noticeable R-D degradation. This indicates that a single learned predictor can generalize to

different (including unseen) stage configurations without retraining, allowing on-the-fly reconfiguration at inference time.

Therefore, we adopt the joint predictor in PACE by default, as it simplifies deployment, maintains a stable model size and memory footprint as $N$ grows, and achieves compression performance nearly identical to that of multiple specialized predictor heads.

*Table 4.* Ablation study on training strategy of stage-scalable predictor

| CONFIG | INDIVIDUAL | | JOINT | |
|---|---|---|---|---|
| | PARAM. ↓ | BD-BR ↓ | PARAM. ↓ | BD-BR ↓ |
| OURS-FP | $10.00M + 0.57M \times N$ | -50.68% | $10.00M + 0.57M \times 1$ | -50.19% |
| OURS-AR | $10.00M + 0.57M \times N$ | -55.93% | $10.00M + 0.57M \times 1$ | -55.07% |

## C. Density-Controlled Experiments

To further examine how point density affects compression performance, we conduct density-controlled experiments on the Ford dataset. Starting from the original point clouds, we apply random downsampling by factors of $2\times$ and $4\times$, retaining 50% and 25% of the original points, respectively. The resulting density levels are visualized in Fig. 2.

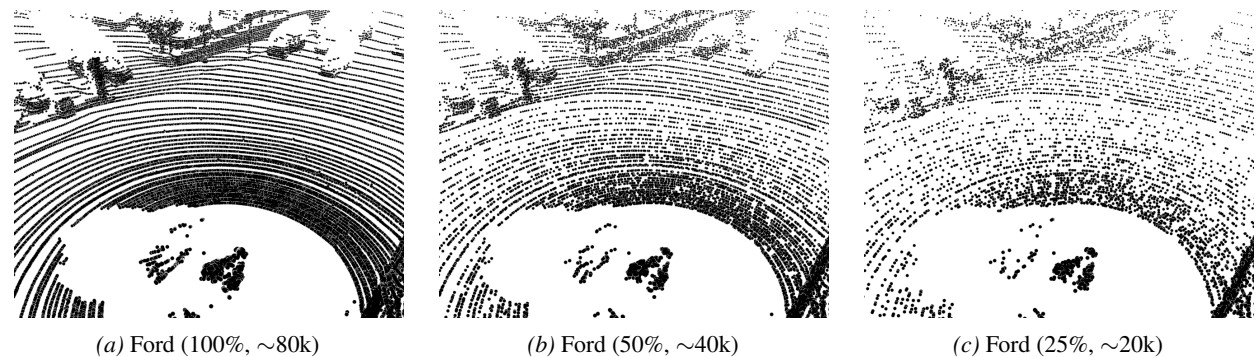

*(a)* Ford (100%, ~80k)          *(b)* Ford (50%, ~40k)          *(c)* Ford (25%, ~20k)

*Figure 2.* Visualization of LiDAR point clouds under different density levels (100%, 50%, and 25%) on the Ford dataset, corresponding to approximately 80k, 40k, and 20k points, respectively.

*Table 5.* BD-BR comparison under controlled point density on the Ford dataset

| METHOD | FORD ($1\times$) | FORD ($2\times$) | FORD ($4\times$) |
|---|---|---|---|
| G-PCC (PREDGEOM) | -45.27% | -34.53% | -29.04% |
| L3C2 | -45.67% | -35.12% | -29.79% |
| OURS-FP | -50.19% | -37.19% | -30.00% |
| OURS-AR | -55.07% | -40.78% | -34.06% |

As reported in Table 5 and illustrated by the R-D curves in Fig. 3, our method achieves the largest gains on the original Ford data. However, as the point density decreases, the performance gap between our method and predtree-based codecs becomes progressively smaller. In particular, the advantage of Ours-FP over L3C2 is clear on the original data but becomes marginal after $4\times$ downsampling. Ours-AR still maintains stronger performance under this controlled Ford setting, but the overall trend indicates that the benefit of learned octree-context modeling is density-dependent.

This behavior is related to the different modeling mechanisms of octree-based and predtree-based codecs. Predtree-based methods, such as G-PCC (Predgeom) and L3C2, encode geometry mainly through prediction links among observed points. When the point cloud becomes sparser, both the number of observed points and the number of prediction links decrease proportionally. In contrast, octree-based methods still need to represent the hierarchical occupancy structure of the space. Although downsampling reduces the number of occupied leaf nodes, the cost of encoding upper-level tree structures does not decrease strictly in proportion to the number of points. Therefore, predtree-based codecs become increasingly competitive on sparse point clouds. This density-controlled study also helps explain our cross-dataset observations: on naturally sparse

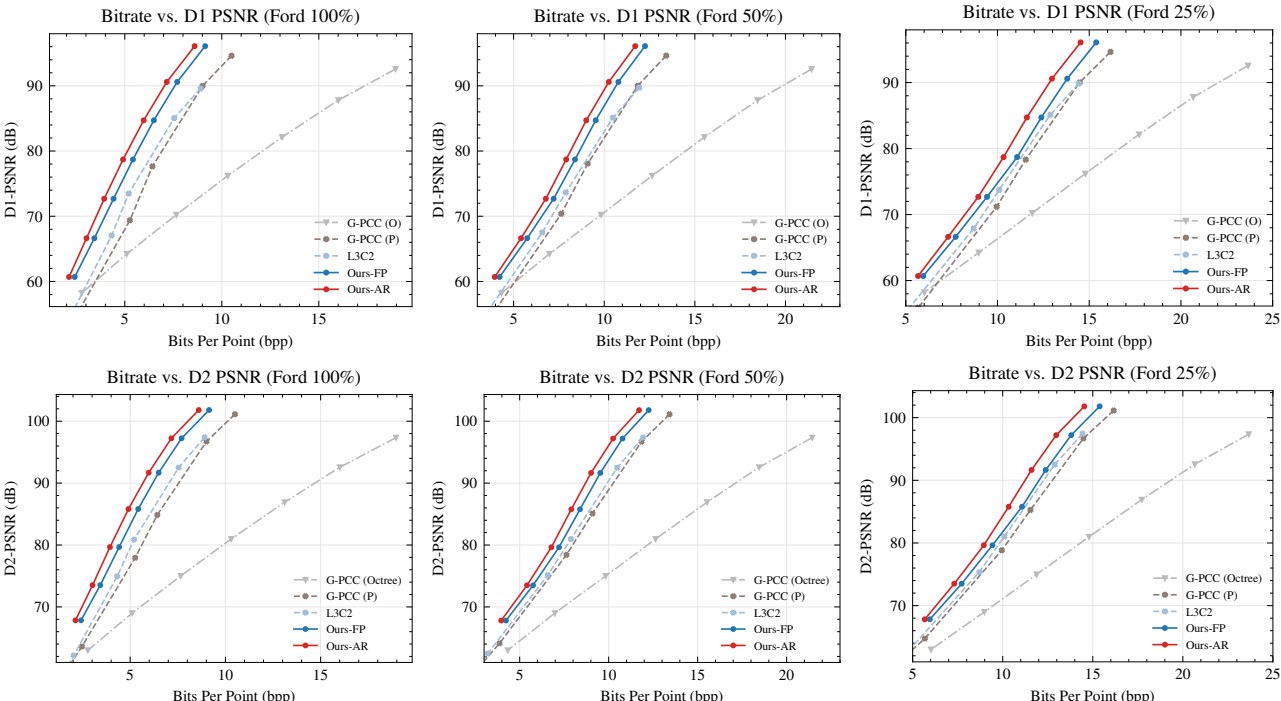

*Figure 3.* Comparison of rate-distortion (R-D) performance under controlled point density on the Ford dataset. We apply random downsampling with three ratios (100%, 50%, and 25%) to simulate varying sparsity levels. G-PCC (O) and G-PCC (P) denote the Octree and Predgeom configurations, respectively.

datasets such as QNX, G-PCC (Predgeom) and L3C2 can outperform our method because their representation is better aligned with sparse geometry. Nevertheless, our approach remains highly competitive and provides consistent baseline improvements.

## D. Special Case of Autoregressive Mode

In our framework, the AR mode is a special case in terms of implementation. For a general $S$-stage decoding strategy, we run the backbone once per window (window length $W = 1024$), and then invoke the stage-scalable predictor once per stage, each time performing full-window inference. A naive AR realization under this pipeline would still require $W$ predictor calls per window, which is computationally prohibitive. In our implementation, a full-window backbone pass takes ∼6 ms, while a single predictor call takes ∼0.3 ms; thus, even with only one backbone execution, $1024\times$ predictor invocations per window lead to excessive latency.

To accelerate AR, we further reduce its overhead by leveraging Mamba's step inference. Specifically, step inference exposes a stateful token-by-token interface: instead of recomputing a full window forward pass for each prediction, the model maintains recurrent states and updates them incrementally so that each step consumes a context of length 1 and produces the next-symbol distribution. This converts AR decoding into a streaming process with stable per-token cost, and the resulting throughput corresponds to the AR speed reported in our paper.

In principle, the encoder could further exploit parallelism because the full octree information is known in advance, allowing the predictor to be evaluated in a vectorized manner (the same as the throughput of the one-stage mode). However, entropy coding requires the encoder and decoder to use exactly the same probability outputs in the same order; even tiny floating-point deviations (e.g., due to GPU non-associativity, different kernel fusion/reduction orders, or nondeterministic execution) can lead to mismatched cumulative frequencies and thus break encoder-decoder synchronization. In our experiments, we observed that such "near-identical" probabilities under parallel encoding can still be unsafe for strict bit-exact arithmetic coding. Therefore, to guarantee bitstream correctness, we enforce a symmetric implementation and use the same step inference path for both encoding and decoding; that is, in AR mode, our encoder processes nodes sequentially rather than in parallel. Consequently, encoding time is relatively long in AR mode.

It is possible to recover safe encoder-side parallelism by further discretizing the probability model, e.g., via quantization or fixed-point implementations with explicit rounding, so that the predicted distributions become bit-exact across execution orders and hardware backends. We leave this direction for future work.

## E. Comparison with Predtree-based Methods

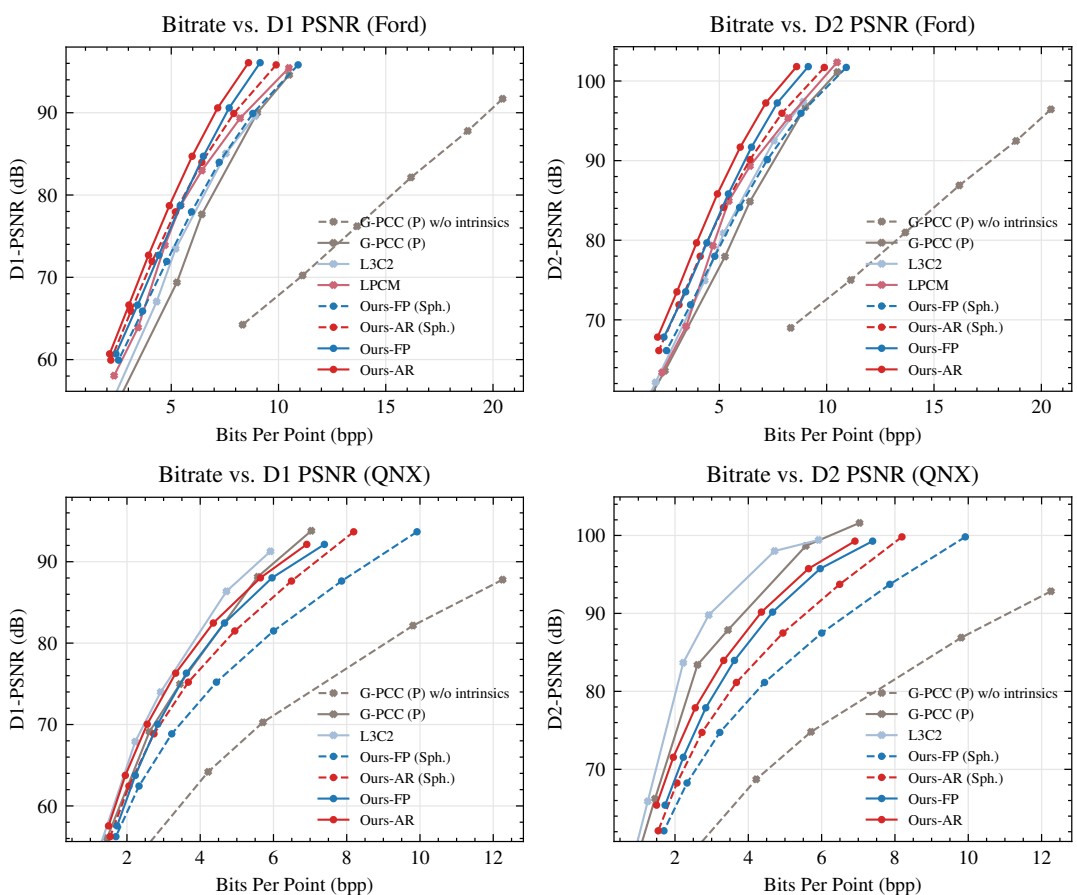

*Figure 4.* Comparison with predtree-based methods on Ford and QNX. G-PCC (P) represents G-PCC (Predgeom) in the figures.

Predtree-based methods, especially rules-based ones, typically rely on accurate LiDAR intrinsics for prediction-tree construction. They are more effective on intrinsics-provided datasets and can outperform octree-based methods under these conditions (e.g., Ford and QNX). However, as shown in Fig. 4, by incorporating intrinsic-aware preprocessing, PACE equips the octree with the ability to leverage intrinsic data, leading to substantial compression improvements in these scenarios.

**Discussion on QNX.** Further, we observe in Fig. 4 that G-PCC (Predgeom) and L3C2 attain slightly higher coding performance than PACE on QNX, even when PACE uses the provided intrinsics (Ours-FP and Ours-AR). This occurs due to the extreme sparsity of QNX (Velodyne VLP16, ~30k points/frame, see Fig. 1d). In such sparse frames, most local 3D neighborhoods contain very few occupied cells, and thus, the local context that a learned octree/voxel-based refinement typically relies on becomes less informative. In contrast, predtree-based codecs model geometry mainly via prediction links among observed points, which can remain effective even when the underlying occupancy neighborhoods are extremely sparse. This analysis is supported by our experimental results on Ford, where the point cloud samples are relatively denser than those in QNX, and PACE outperforms both G-PCC (Predgeom) and L3C2 remarkably.

**Discussion on Intrinsics.** To further examine the reliance on intrinsics and the resulting cross-dataset portability, we additionally re-evaluate G-PCC (Predgeom) by removing the intrinsics from Ford and QNX (G-PCC (P) w/o intrinsics in Fig. 4). Without intrinsics, G-PCC (Predgeom) degrades notably, while PACE remains robust (see Ours-FP (Sph.) and Ours-AR (Sph.)); moreover, L3C2 fails to operate under the intrinsics-removed setting. These results indicate that the

performance of predtree-based codecs is tightly coupled to the availability of precise intrinsics, whereas PACE offers wide generality and compatibility even when such metadata is missing or unreliable.

For readers' convenience, we capture the configuration of G-PCC (Predgeom) and L3C2 in the gray rectangles of this supplementary material to demonstrate the sensor intrinsics required by them.

```
// Ford
numLasers: 64

lasersTheta:
    -0.461611, -0.451281, -0.440090, -0.430000, -0.418945, -0.408667, -0.398230, -0.388220,
    -0.377890, -0.367720, -0.357393, -0.347628, -0.337549, -0.327694, -0.317849, -0.308124,
    -0.298358, -0.289066, -0.279139, -0.269655, -0.260049, -0.250622, -0.241152, -0.231731,
    -0.222362, -0.213039, -0.203702, -0.194415, -0.185154, -0.175909, -0.166688, -0.157484,
    -0.149826, -0.143746, -0.137673, -0.131631, -0.125582, -0.119557, -0.113538, -0.107534,
    -0.101530, -0.095548, -0.089562, -0.083590, -0.077623, -0.071665, -0.065708, -0.059758,
    -0.053810, -0.047868, -0.041931, -0.035993, -0.030061, -0.024124, -0.018193, -0.012259,
    -0.006324, -0.000393,  0.005547,  0.011485,  0.017431,  0.023376,  0.029328,  0.035285

lasersZ:
    29.900000, 26.600000, 28.300000, 24.600000, 26.800000, 25.100000, 24.800000, 22.400000,
    22.400000, 21.900000, 23.000000, 20.700000, 21.100000, 20.300000, 19.900000, 19.000000,
    18.900000, 15.300000, 17.300000, 16.000000, 16.200000, 15.100000, 14.800000, 14.400000,
    13.800000, 13.000000, 12.700000, 12.100000, 11.500000, 11.000000, 10.400000,  9.800000,
    10.700000, 10.300000, 10.000000,  9.400000,  9.100000,  8.600000,  8.200000,  7.700000,
     7.400000,  6.800000,  6.500000,  6.000000,  5.600000,  5.100000,  4.700000,  4.300000,
     3.900000,  3.500000,  3.000000,  2.600000,  2.100000,  1.800000,  1.300000,  0.900000,
     0.500000, -0.100000, -0.400000, -0.900000, -1.200000, -1.700000, -2.100000, -2.500000

lasersNumPhiPerTurn:
    800, 800, 800, 800, 800, 800, 800, 800, 800, 800, 800, 800, 800, 800, 800, 800,
    800, 800, 800, 800, 800, 800, 800, 800, 800, 800, 800, 800, 800, 800, 800, 800,
    4000, 4000, 4000, 4000, 4000, 4000, 4000, 4000, 4000, 4000, 4000, 4000, 4000, 4000, 4000, 4000,
    4000, 4000, 4000, 4000, 4000, 4000, 4000, 4000, 4000, 4000, 4000, 4000, 4000, 4000, 4000, 4000
```

```
// QNX
numLasers: 16

lasersTheta:
    -0.268099, -0.230939, -0.194419, -0.158398, -0.122788, -0.087491, -0.052410, -0.017455,
     0.017456,  0.052408,  0.087487,  0.122781,  0.158381,  0.194378,  0.230865,  0.267953

lasersZ:
    -2.000000, -1.500000, -1.300000, -1.100000, -1.000000, -1.000000, -1.000000, -1.000000,
     0.000000,  0.000000, -0.100000, -0.200000, -0.200000, -0.200000, -0.300000, -0.200000

lasersNumPhiPerTurn:
    360, 360, 360, 360, 360, 360, 360, 360, 360, 360, 360, 360, 360, 360, 360, 360
```

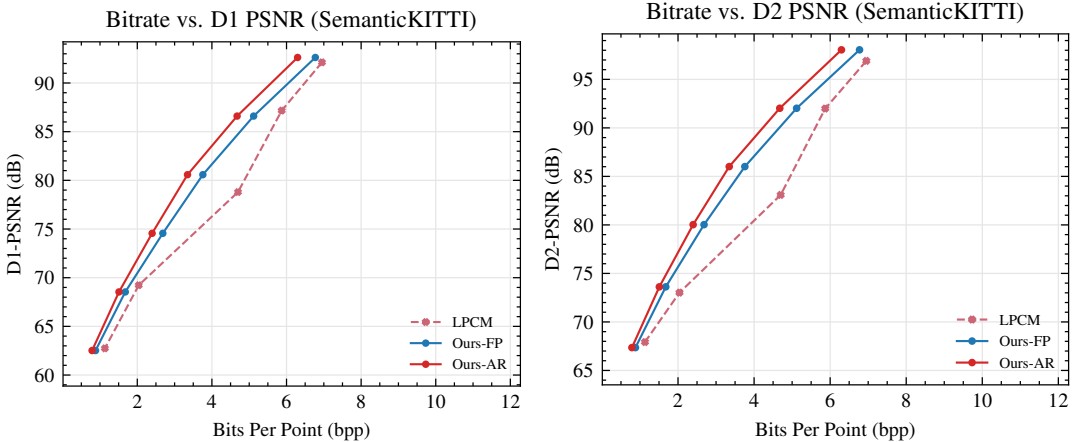

*Figure 5.* Comparison with learning-based predtree method LPCM on SemanticKITTI.

**Comparison with LPCM.** LPCM is a recent learning-based predtree method built upon classic rules-based predtree codecs. It does not strictly require intrinsics and can be applied to datasets without sensor intrinsics such as SemanticKITTI. Nevertheless, when reliable intrinsics are unavailable, the geometric priors used to form the prediction tree become substantially weaker, which can result in a dramatic performance drop. As shown in Fig. 5, LPCM performs much worse on SemanticKITTI than on Ford, where intrinsics are provided.

Notably, LPCM adopts a two-branch design to handle different bitrates. As evidenced by the R-D curves, it exhibits a clear breakpoint, which is induced by switching between two coding branches between high- and low-bitrates. At high bitrates, LPCM follows a predtree-style predictive coding pipeline. At low bitrates, however, LPCM switches to an autoencoder branch. This is because when the underlying geometric priors are weak or noisy (e.g., under missing/unreliable intrinsics), predictive coding becomes brittle at low bitrates, and the overall R–D behavior may deteriorate sharply.

By contrast, our PACE adopts a monolithic solution to cover a wide bitrate range. It can be observed from Fig. 4 and Fig. 5 that PACE consistently outperforms LPCM at all bitrate points on both Ford and SemanticKITTI. Since LPCM only provides results on these two datasets, we cannot compare with it on additional datasets such as QNX and nuScenes.

## F. Qualitative Results

We visualize the LiDAR point cloud reconstructed by different methods in Fig. 6 for subjective quality assessment.

## G. Limitations

As shown in Table 4 of the main manuscript and Tables 2 and 3 of this supplementary material, the encoding time of our autoregressive mode is much longer than that of other modes. This speed could be further improved, as all required information is available at the encoder and parallel processing is feasible. However, as discussed in Section D, implementing such parallelism would require techniques such as quantization and hardware optimization, which are beyond the scope of this work. Future work will focus on optimizing PACE for faster encoding.

Currently, PACE does not meet real-time processing requirements. Future work will focus on accelerating its coding speed through efficient attention networks, model distillation, and other techniques, while preserving compression performance as much as possible.

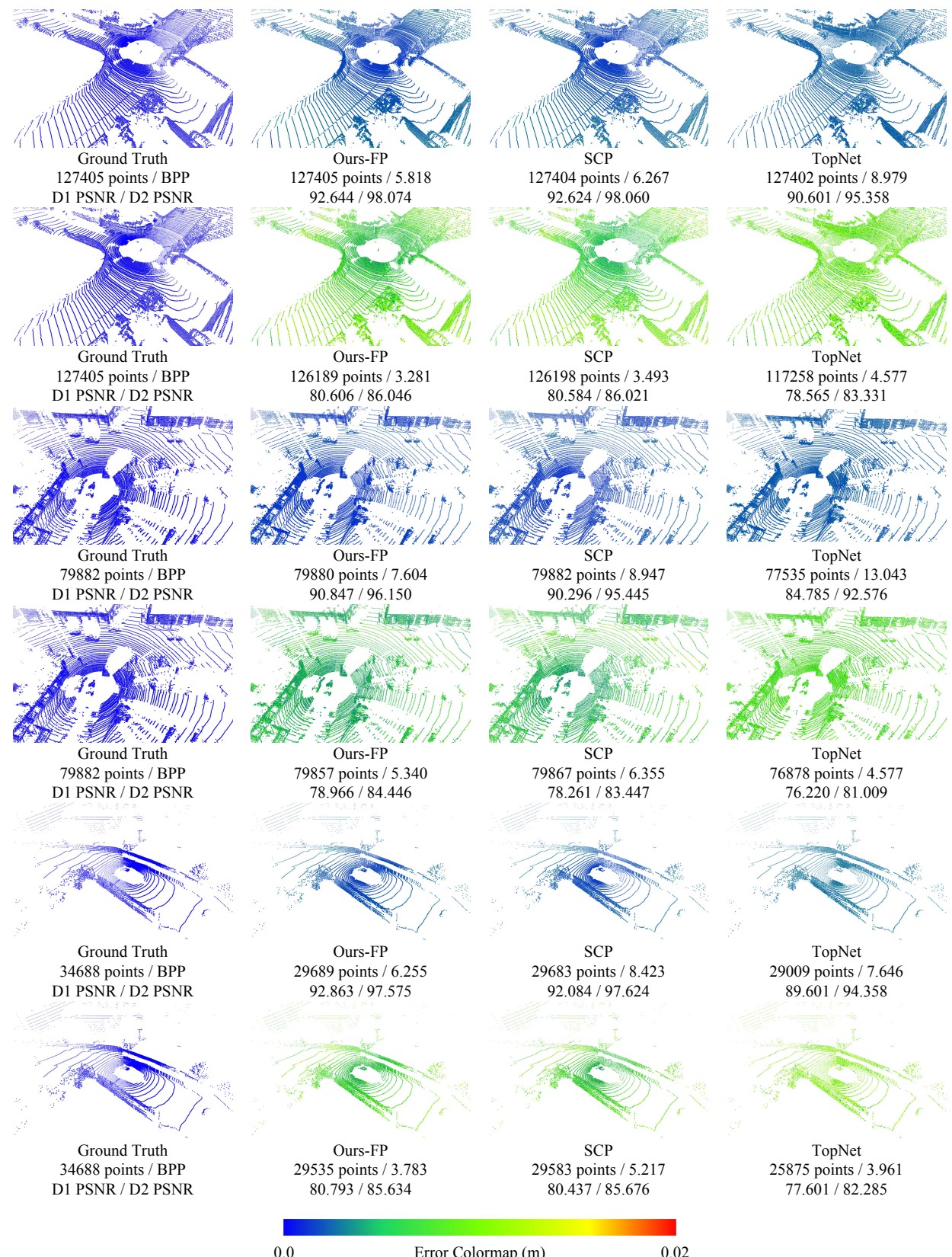

*Figure 6.* Qualitative visualization. Rows 1, 3, and 5 present compression results at higher bitrates, while Rows 2, 4, and 6 show results at lower bitrates. Representative samples from SemanticKITTI (Rows 1-2), Ford (Rows 3-4), and nuScenes (Rows 5-6) are shown.

