# OpenReview forum: "PACE: Post-Causal Entropy Modeling for Learned LiDAR Point Cloud Compression"
_ICML.cc/2026/Conference — ICML 2026 regular_

### Official Review · Reviewer_UxVL · 2026-02-26

**Soundness:** 3
**Presentation:** 3
**Significance:** 3
**Originality:** 3
**Overall Recommendation:** 4
**Confidence:** 2

**Summary:**

The paper is motivated by the severe bottleneck introduced by causal octree decoding in learned point cloud compression pipelines. To address this, it proposes a non-causal backbone that decouples feature extraction from the causal system via point/token voxelization and a lookup-table–based association mechanism for efficient token-to-node mapping. Experiments report improved throughput/latency while maintaining competitive performance across SemanticKITTI, nuScenes, Ford, and QNX.

**Compliance With Llm Reviewing Policy:**

Affirmed.

**Final Justification:**

All of my concerns have been addressed, and I now lean toward acceptance. Therefore, I am raising my score.

**Key Questions For Authors:**

Please address the concerns listed in Weaknesses.

**Limitations:**

yes

**Strengths And Weaknesses:**

### **Strengths**
- The problem setting and motivation are clearly articulated, and the paper targets a practically important bottleneck in octree-based point cloud compression.
- The idea of decoupling feature extraction/backbone design from the causal octree decoding process is conceptually novel and addresses a key limitation of prior causal pipelines.
- The non-causal backbone design (e.g., tokenization with a lookup table-based association mechanism) appears technically sound and implementation-friendly, with a clear path to improving parallelism and throughput.

### **Weaknesses**
- As octree depth L increases (i.e., finer spatial resolution) or scenes become more complex, the number of occupied nodes and tokens can grow rapidly, increasing memory footprint and runtime. The paper does not adequately characterize how bitrate/latency and GPU memory scale with L and token count, nor does it discuss potential failure modes (e.g., throughput collapse) in high-resolution settings.
- The analysis of why the method underperforms on sparse point clouds (e.g., QNX) is incomplete. The manuscript would be stronger with density-controlled experiments (e.g., uniform downsampling or voxel-grid thinning) to explicitly track compression performance as point density decreases.
- The justification for fair comparison is unclear. The experiments use different voxel/geometry precisions across datasets (e.g., Ford/QNX fixed at 1 mm, while SemanticKITTI and nuScenes use dataset-specific scaling factors), without explaining why precision is not unified (e.g., 1 mm) or matched at equal effective precision/bitrate. This raises the concern that the reported gains may partly reflect dataset-specific quantization choices rather than the proposed method.

---

> ### Author Rebuttal · Authors · 2026-03-31
>
> We sincerely appreciate your constructive feedback. We are greatly encouraged by your recognition that our paper **targets a practically important bottleneck** and offers a **conceptually novel**, **implementation-friendly** solution. We highly value this opportunity to address your remaining concerns.
>
> **1. Scalability and Physical Limits**
>
> *1.1 Scaling with Octree Depth*
>
> We have carefully evaluated the scalability of PACE across octree depths from $L$=11 to $L$=17. As shown in the tables below, PACE scales linearly with increasing depth. Notably, it maintains a highly stable GPU memory footprint, yielding only a marginal increase from 0.54 GB to 0.60 GB. This minimal memory overhead underscores the efficiency of PACE for processing high-resolution point clouds.
>
> Computational complexity of Ours-FP across different octree depths on the Ford dataset:
>
> |Depth|#Token (k)|Bpp|Enc. (s)|Dec. (s)|Mem. (GB)|
> |:-|-:|-:|-:|-:|-:|
> |11|107|2.42|0.72|0.60|0.54|
> |13|252|4.42|1.33|1.16|0.57|
> |15|419|6.50|2.06|1.92|0.59|
> |17|587|9.14|2.78|2.63|0.60|
>
> Computational complexity of Ours-AR across different octree depths on the Ford dataset:
>
> |Depth|#Token (k)|Bpp|Enc. (s)|Dec. (s)|Mem. (GB)|
> |:-|-:|-:|-:|-:|-:|
> |11|107|2.13|6.68|9.84|0.54|
> |13|252|3.94|9.31|14.39|0.57|
> |15|419|5.97|12.19|17.63|0.59|
> |17|587|8.60|14.88|21.68|0.60|
>
> *1.2 Potential Failure Modes*
>
> PACE is explicitly designed to mitigate bottlenecks in extreme settings. By partitioning nodes into disjoint windows of size $W$=1024, our unmasked multihead attention is strictly bounded by $O(W^2)$ within each window. Thus, as $L$ increases, our memory footprint scales linearly with the number of windows, effectively preventing out-of-memory (OOM) failures. In addition, PACE isolates context aggregation into a non-causal backbone executed only once per window, eliminating repetitive backbone executions and allowing throughput to degrade smoothly rather than rapidly.
>
> **2. Density-Controlled Experiments**
>
> Following your suggestion, we conducted comparative experiments where we strictly controlled the point cloud density.
>
> Specifically, we applied random downsampling to the Ford dataset by factors of 2 and 4 (denoted as 2x and 4x, respectively). As shown in the table below, our FP model achieves its largest performance gains on the original Ford data; however, as the point cloud density decreases, its relative advantage over L3C2 gradually diminishes.
>
> BD-BR gains compared to G-PCC (octree) anchor on the Ford dataset:
>
> |Model|Ford (1x)|Ford (2x)|Ford (4x)|
> |:-|-:|-:|-:|
> |G-PCC (P)|-45.27%|-34.53%|-29.04%|
> |L3C2|-45.67%|-35.12%|-29.79%|
> |Ours-FP|-50.19%|-37.19%|-30.00%|
> |Ours-AR|-55.07%|-40.78%|-34.06%|
>
> This shift can be attributed to the fundamental differences in how these codecs model geometry. Predtree-based codecs (e.g., L3C2) model geometry mainly via prediction links among observed points. When the point count is reduced by a factor of 4, the number of links to be encoded decreases proportionally. In contrast, due to the hierarchical structure of octree-based methods, while the number of occupied leaf nodes drops, the structural overhead of encoding the upper-level tree nodes does not scale down linearly with the point count. As a result, predtree methods naturally excel on highly sparse data; nevertheless, our approach remains highly competitive and provides consistent baseline improvements.
>
> Visualizations of downsampled scans and rate-distortion curves are available at this [anonymous link](https://anonymous.4open.science/r/icml_8078).
>
> **3. Clarification of Quantization Settings**
>
> We would like to provide a further explanation of our experimental settings:
>
> - **Ford and QNX** are MPEG-defined standard test sequences that have been pre-quantized to 1 mm precision (18-bit). We strictly follow the MPEG G-PCC common test conditions (CTC) [1], ensuring full consistency with prior works under the same benchmark.
>
> - **SemanticKITTI** follows the quantization strategy adopted in prior works (e.g., EHEM, SCP, and ASRL) to ensure a fair comparison. Specifically, these works adopt the form $Q/(2^L−1)$ with Q=400, which maps the sensing range (<80 m) to a fixed octree depth, and we use the same setting in our experiments.
>
> - **nuScenes** is an additional dataset that we include to evaluate model robustness. To make the results comparable, we adopt a consistent quantization strategy aligned with SemanticKITTI. Given that nuScenes has a slightly larger sensing range (~100 m), the parameter Q is increased to 450 to ensure that all frames are properly quantized at the same octree depth. Crucially, all baseline models are retrained under the same quantization settings.
>
> We hope that these explanations address your concerns. We will carefully integrate these new results, ablation studies, and clarifications into the manuscript and the appendix to ensure a comprehensive yet concise final paper.
>
> [1] WG 07. Common test conditions for G-PCC. Output document N00944, 2024.

---

> > ### Author Rebuttal · Reviewer_UxVL · 2026-04-01
> >
> > As the authors’ response has satisfactorily addressed my concerns, I have decided to raise my score.

---

### Official Review · Reviewer_2TYR · 2026-03-09

**Soundness:** 4
**Presentation:** 3
**Significance:** 3
**Originality:** 3
**Overall Recommendation:** 5
**Confidence:** 2

**Summary:**

The paper proposes PACE, a post-causal entropy modeling framework for LiDAR point cloud compression that decouples context aggregation from causal prediction, enabling flexible stage-wise decoding while reducing redundant backbone computation. The method demonstrates strong compression performance and substantial decoding speed improvements across multiple datasets.

**Compliance With Llm Reviewing Policy:**

Affirmed.

**Key Questions For Authors:**

# Questions
**Sensitivity to backbone design.**
The model incorporates several architectural components, attention layers, and a Mamba-based sequence model. It would be helpful to understand how sensitive the overall performance is to these design choices. For instance, could a simpler backbone or an alternative sequence model achieve similar improvements?

**Encoding runtime.**
Although the method substantially reduces decoding latency, the encoding time in the autoregressive setting appears relatively high compared to the fully parallel configuration. It would be useful to clarify whether encoding latency could become a bottleneck in real-time systems, and whether there are potential strategies to further parallelize the encoding process.

**Hardware deployment considerations.**
The paper emphasizes practical deployment in autonomous systems. However, it is unclear whether the method has been evaluated on edge hardware or embedded platforms commonly used in autonomous driving pipelines.

**Limitations:**

Yes, the author have discussed about the limitation in both appendix and the main paper.

**Strengths And Weaknesses:**

# Strengths



1. **Clean and intuitive design.**
The idea of post-causal modeling separates context aggregation from causal prediction makes sense to me. This design is intuitive and helps reduce redundant computation during decoding.

2. **Flexible stage-scalable predictor.**
The predictor supports multiple decoding modes (parallel, multi-stage, autoregressive) within the same model, allowing flexible trade-offs between compression performance and runtime latency.

3. **Strong empirical results.**
Experiments on several LiDAR datasets (SemanticKITTI, Ford, nuScenes, QNX) show consistent BD-BR improvements and significantly faster decoding.


# Weaknesses
Overall, I did not observe any major weaknesses in this work. Instead, I include several questions regarding minor limitations and clarifications, which are listed in the questions section.

---

> ### Author Rebuttal · Authors · 2026-03-31
>
> We sincerely appreciate your recognition of our work's **clean and intuitive design** and **strong empirical results**. We welcome this opportunity to clarify your remaining questions.
>
> **1. Architectural Sensitivity**
>
> The core post-causal framework of PACE is fundamentally architecture-agnostic. While specific modules govern the capacity-complexity trade-off, the framework's advantages do not rely on singular design choices. We demonstrate this robustness across its two primary structural components:
>
> - In our current implementation, we adopt a standard 12-layer Attention backbone as a straightforward baseline. To evaluate architectural sensitivity, we conducted experiments on the Ford dataset using a reduced 3-layer variant. Under this setting, performance decreased from -50.19% to -46.24% in FP mode. This ~4% difference demonstrates that while a higher-capacity backbone yields peak performance, the framework maintains significant compression gains even with a substantially simplified design.
>
> - While we utilize Mamba in the predictor for its superior efficiency, this choice is not mandatory; alternative autoregressive models can be readily integrated. As detailed in our response to Reviewer KFTA (see "Necessity of the Mamba Module" section), replacing Mamba with Attention achieves highly comparable compression performance (<1% difference), although this incurs over $2\times$ the computational complexity.
>
> In summary, while individual components modulate the final efficiency-accuracy profile, PACE's core contributions of *eliminating repeated backbone execution and enabling flexible inference* are inherent to the proposed paradigm itself.
>
> **2. Autoregressive Encoding Acceleration**
>
> We agree that the current encoding latency in the autoregressive (AR) mode is a bottleneck for real-time systems, and we appreciate the opportunity to clarify the theoretical and practical nuances of this limitation.
>
> **The Bit-Exactness Bottleneck**. Theoretically, as the encoder has a priori access to the complete ground-truth point cloud, it could process the entire window in parallel. This would reduce the AR encoding time to match our fully parallel (FP) mode. However, entropy coding strictly requires bit-exact probability distributions between the encoder and decoder. Due to the non-associativity of floating-point arithmetic, parallel execution at the encoder and serial execution at the decoder introduce numerical discrepancies. Even minute probability mismatches cause the arithmetic coder to desynchronize and fail. Thus, to guarantee correctness, our current prototype constrains the encoder to mirror the decoder's serial execution path.
>
> **Future Optimization**. Ensuring bit-exactness across different execution paths and platforms remains an open challenge for AI-based codecs, as highlighted in MPEG standardization efforts [1][2]. A proven solution [3] utilizes *model quantization* to avoid floating-point calculations. Under fixed-point arithmetic, operations become strictly associative, safely decoupling a fast parallel encoder from a serial decoder. As implementing quantization for our model entails substantial engineering effort, we leave this optimization for future work.
>
> [1] WG 07. Towards reproducible learning-based compression. MPEG input m70284, 2024.
>
> [2] WG 07. Cross-platform reproducibility. MPEG input m71396, 2025.
>
> [3] Yu J, et al. Mixed-precision post-training quantization for learned image compression. IEEE IoTJ, 2025.
>
> **3. Hardware Deployment**
>
> We thank the reviewer for highlighting the importance of edge hardware evaluation. To demonstrate practical viability, we deployed PACE on an NVIDIA Jetson Thor edge device for this rebuttal.
>
> Given the computational constraints of edge devices, the 3-layer variant is employed. As detailed in the "Architectural Sensitivity" section in this response, this lightweight model incurs only about 4% compression penalty. Experiments demonstrate that the model (FP mode) achieves an encoding speed of 0.35 s/frame on the NVIDIA Thor for 12-bit QNX point clouds (where 12-bit yields a precision of ±3.2 cm, which is sufficiently accurate for detection [4]). For 10-bit precision, the speed further improves to 0.15 s/frame (where 10-bit yields a precision of ±12.8 cm, adequate for SLAM tasks [5]). We believe further engineering optimizations (e.g., INT8 quantization and TensorRT deployment) will fully unlock the efficiency of the model for autonomous systems.
>
> [4] You K, et al. RENO: Real-time neural compression for 3D LiDAR point clouds. CVPR, 2025.
>
> [5] Xu W, et al. Fast-LIO2: Fast direct LiDAR-inertial odometry. IEEE ToR, 2022.
>
> Thank you again for your feedback. We hope this response resolves your questions, and we will ensure these experiments and analyses are included in the final version of the paper and supplementary material.

---

> > ### Author Rebuttal · Reviewer_2TYR · 2026-04-04
> >
> > All my concerns are well addressed, and I will keep my score.

---

### Official Review · Reviewer_KFTA · 2026-03-10

**Soundness:** 3
**Presentation:** 3
**Significance:** 3
**Originality:** 3
**Overall Recommendation:** 4
**Confidence:** 3

**Summary:**

Targeting the octree-based LiDAR point cloud compression (LPCC) task, this paper proposes a novel framework named PACE (Post-Causal Entropy modeling). PACE decouples the inter-level (ancestor) context extraction into a non-causal backbone that only needs to be executed once, and restricts the intra-level causal dependencies to a lightweight, stage-scalable predictor. Combined with elastic causal embedding (ECE) and Mamba-based distribution estimation, a single PACE model can support an arbitrary number of stage partitions, ranging from fully parallel (one-stage) to fully autoregressive.

**Compliance With Llm Reviewing Policy:**

Affirmed.

**Final Justification:**

All my concerns are well addressed, and I will keep my score.

**Key Questions For Authors:**

1) Regarding the potential solution for AR mode encoding latency: Appendix C mentions that encoder-side parallelism could be recovered in the future via quantization or fixed-point implementations to avoid arithmetic coding desynchronization caused by floating-point errors. Could you please clarify: in your current evaluations, if full-window parallel prediction is forced on the encoder side, how frequently do floating-point errors actually occur and lead to decoding crashes?

2) Necessity of the Mamba Module: The stage-scalable predictor incorporates a Mamba-based Selective State Space Model (SSM) for distribution estimation. Compared to the more commonly used lightweight Transformers or MLPs, what are the specific advantages of employing Mamba in this context?

**Limitations:**

yes

**Strengths And Weaknesses:**

Strengths:
1) The paper provides a profound analysis of the bottlenecks in existing works, attributing the repetitive computation issue to the tight coupling between the backbone and the prediction head in fully-causal models (e.g., TopNet). It subsequently proposes a post-causal decoupling design, presented with smooth writing and clear logic.

2) The proposed stage-scalable predictor achieves a one-size-fits-all capability.

3) Evaluated across four representative mainstream datasets, PACE outperforms existing learned octree-based models as well as the traditional G-PCC standard in terms of both performance and efficiency.


Weaknesses:
1) Issue of Overclaiming: The introduction and conclusion sections prominently claim that the model is highly attractive for practical applications and repeatedly emphasize the dramatic reduction in decoding latency. However, it is only disclosed in Table 2 of the main text and Appendix F that the encoding time for its autoregressive (AR) mode is as high as 10.89 seconds. For autonomous driving scenarios that demand strict real-time performance, this represents a fatal shortcoming.

2) Inaccurate Use of the Term "Non-Causal": The paper designates the backbone responsible for processing inter-level context as a "non-causal backbone." However, in reality, this network still strictly relies on ancestor node information from previous levels (inter-level), which inherently makes it a cross-level causal system. The "non-causality" referred to here is strictly confined to intra-level siblings. The usage of this term is overly broad and prone to causing misunderstandings.

---

> ### Author Rebuttal · Authors · 2026-03-31
>
> We sincerely appreciate your time and thoughtful comments. We highly value your positive remarks regarding our **profound analysis of the bottlenecks in existing works** and the presentation of **a novel framework**, as well as our **smooth writing and clear logic**. We welcome this opportunity to address your remaining concerns.
>
> **1. Clarification on Practicality and Latency Claims**
>
> We appreciate this feedback and will tone down these claims in the revision. To clarify, the AR mode serves only as a theoretical performance upper bound. As demonstrated in Table 4 of the manuscript, intermediate configurations (e.g., 8-stage or 16-stage) already achieve a strong trade-off, delivering most of the compression gains of AR while maintaining near-FP runtime. Additionally, our "low-latency" claim refers strictly to PACE's avoidance of the repeated backbone executions of prior AR/multi-stage methods, significantly reducing decoding costs. We will make this context explicit.
>
> **2. Refining the Terminology for the Backbone**
>
> We fully agree that because the backbone relies on inter-level ancestor nodes, it remains inherently causal across octree levels. Our original use of "non-causal" was solely intended to describe the removal of sequential constraints among intra-level siblings. While we did attempt to bound our scope to "intra-level stage-wise causality" in Section 3.2, we agree with you that using "non-causal" to name the backbone is overly broad. We will revise the term to a more precise alternative to avoid potential misunderstanding.
>
> **3. Explanation of Floating-point Errors**
>
> In our empirical evaluations, we observed that *if full-window parallel prediction is forced on the encoder side, the decoding crash rate is nearly 100% for realistically sized point clouds.*
>
> The root cause is that floating-point arithmetic does not satisfy the associative law. When the GPU executes across different computational paths, small numerical discrepancies (typically on the order of 1e-6) can arise from differences in accumulation order. While such deviations are negligible for most learning tasks, they are problematic in entropy coding. As discussed in Appendix C, arithmetic coders require encoder and decoder to maintain identical cumulative frequency tables at each symbol; a single discrepancy anywhere in the sequence will corrupt the entire subsequent bitstream, leading to decoding failure. Given that a typical point cloud contains hundreds of thousands of nodes, it is virtually certain that at least one such discrepancy will occur.
>
> For this reason, we enforce a symmetric step-inference path for both encoder and decoder in AR mode, which guarantees decoding correctness at the cost of longer AR encoding time. **Ensuring bit-exact reproducibility** across different execution paths and hardware platforms remains a critical open challenge for AI-based codecs, as evidenced by its **increasing prominence in recent standardization efforts such as MPEG AI-PCC [1][2].**
>
> [1] WG 07. Towards reproducible learning-based compression. MPEG input m70284, 2024.
>
> [2] WG 07. Cross-platform reproducibility. MPEG input m71396, 2025.
>
> **4. Necessity of the Mamba Module**
>
> We would like to clarify that our stage-scalable predictor is not strictly tied to a specific architecture; any sequence model can be utilized for context aggregation before distribution estimation. However, we adopted Mamba as our default instantiation because it achieves the optimal balance between computational efficiency and context modeling.
>
> To demonstrate this, we compare Mamba against a lightweight MLP and an Attention-based predictor in AR mode on the Ford dataset (using G-PCC octree as the anchor for BD-BR calculation):
>
> |Config|Mem. (GB)|Enc. (s)|Dec. (s)|BD-BR (%)|
> |:-|-:|-:|-:|-:|
> |MLPs|0.58|5.04|8.18|-52.49|
> |ATTN|2.68|33.34|40.95|-54.83|
> |MAMBA|0.58|10.89|16.36|-55.07|
>
> As shown in the table:
>
> - Compared to MLPs: Under our elastic causal embedding, MLPs capture only restricted local context (e.g., the immediately preceding token). This limits their ability to model complex conditional distributions for entropy coding, leading to a performance drop.
>
> - Compared to Attention: While Attention captures context effectively, its autoregressive execution suffers from severe computational bottlenecks. Even with a lightweight, single-layer implementation, sequential AR coding results in significantly slower speeds and a larger memory footprint.
>
> Ultimately, Mamba achieves the robust context modeling capability typical of Transformers while maintaining the lightweight, constant-memory footprint of MLPs.
>
> We hope these explanations resolve your concerns. We will carefully refine our manuscript based on your insightful suggestions to ensure a clearer and more rigorously presented paper.

---

> > ### Author Rebuttal · Reviewer_KFTA · 2026-04-03
> >
> > All my concerns are well addressed, and I will keep my score.

---

### Official Review · Reviewer_Un7J · 2026-03-19

**Soundness:** 3
**Presentation:** 3
**Significance:** 3
**Originality:** 3
**Overall Recommendation:** 4
**Confidence:** 2

**Summary:**

This paper proposes a post-causal entropy modeling framework, PACE, to address issues in octree-based LiDAR point cloud compression, such as high decoding latency, rigid trade-offs between performance and latency, and tight coupling between the context backbone and the prediction module. The method designs hierarchical context aggregation as a non-causal backbone network, restricting causal constraints to a lightweight and dynamically scalable predictor. This avoids the substantial computational overhead of repeatedly executing large models and enables support for arbitrary-stage switching with a single model through elastic causal embedding. Additionally, the authors introduce sensor-aware preprocessing and graph-based position encoding to further enhance compression performance, and conduct comprehensive experimental validation. Results show that PACE achieves state-of-the-art compression rates across multiple datasets and reduces decoding latency by over 90% in autoregressive mode, demonstrating high practical value.

**Compliance With Llm Reviewing Policy:**

Affirmed.

**Final Justification:**

I have read the rebuttal, and decide to maintain my score.

**Key Questions For Authors:**

1. What are the feature fusion method and specific implementation details between the non-causal backbone and the causal predictor?

2. What is the specific structure, dimension design, and training strategy of Elastic Causal Embedding (ECE)?

3. Regarding the slow encoding speed in autoregressive mode, what are the immediately implementable optimization strategies within the existing framework?

4. How are the hyperparameters for k-NN graph construction and neighborhood aggregation in Graph Position Encoding (GPE) selected?

5. How is the training process for a single model supporting arbitrary stage switching implemented? Does it employ a multi-stage mixed training strategy?

**Limitations:**

It is suggested that the authors supplement the following content:

The encoding time remains relatively long in autoregressive mode, with the authors only providing an analysis of the causes without offering feasible optimization solutions.

Some module details (such as the implementation of elastic causal embedding and the window partitioning strategy) are described rather concisely, which hinders full reproducibility.

A more comprehensive comparison with the latest concurrent point cloud compression methods is lacking.
The analysis of limitations in extremely sparse scenarios is not sufficiently in-depth.

**Strengths And Weaknesses:**

The problem motivation is clear and practical: it precisely identifies the core bottleneck of existing learning-based octree point cloud compression, namely that fully causal modeling leads to extremely slow decoding and an inability to dynamically balance performance and speed, aligning well with the demands of real-time systems such as autonomous driving and robotics.
The technical design is comprehensive: modules such as sensor-aware preprocessing, graph-based position encoding, elastic causal embedding, and Mamba sequence modeling are well-coordinated and logically coherent.
The experiments are thorough and credible: comparative, ablation, and complexity analyses are conducted on four mainstream datasets, with quantitative support for both compression performance and speed improvements, ensuring strong reproducibility.

---

> ### Author Rebuttal · Authors · 2026-03-31
>
> We sincerely appreciate your thoughtful feedback. Thank you for recognizing our **clear motivation**, **comprehensive technical design**, and **thorough experiments**. We highly value this opportunity to address your concerns.
>
> **1. Implementation Details**
>
> For clarity, we detail our *stage-scalable predictor* using Pytorch-style pseudo code below. As shown, the features generated by the backbone are fused with the current-level node embeddings via element-wise addition. ECE structure and dimensional configurations are also delineated.
>
> ```
> # B: batch size, W: window length, C: channels
> # Default: B=32, W=1024, C=256
>
> # Layer definition:
> self.emb = nn.Embedding(255, C)
> self.ssm = MambaBlocks(C)
> self.pred = nn.Sequential(nn.Linear(C, C),nn.GELU(),nn.Linear(C, 255),nn.Softmax(dim=-1))
>
> def forward(self, feats, nodes, n_stages):
>     """
>     :param feats (FloatTensor): Contextual features from the backbone, shape (B, W, C)
>     :param nodes (LongTensor): Nodes from the current level, shape (B, W)
>     :param n_stages (int): Number of prediction stages
>     :return prob (FloatTensor): Probability distribution for arithmetic coding, shape (B, W, 255)
>     """
>     # One-stage mode
>     if n_stages == 1:
>         return self.pred(self.ssm(feats)) # shape (B, W, 255)
>
>     # Multi-stage mode
>     prob = torch.zeros([B, W, 255])
>     stage_indices = torch.arange(W) % n_stages
>     nodes_emb = self.emb(nodes) # shape (B, W, C)
>
>     # Stage-wise decomposition
>     for stage_idx in range(n_stages):
>
>         siblings_emb = nodes_emb.clone() # shape (B, W, C)
>
>         # ECE: Mask the current and future stages
>         siblings_emb[:, stage_indices >= stage_idx] = 0
>         siblings_emb = F.pad(siblings_emb[:, :-1, :], (0, 0, 1, 0))
>
>         # ECE: Addition
>         fused_feats = feats + siblings_emb # shape (B, W, C)
>
>         # Distribution estimation
>         stage_prob = self.pred(self.ssm(fused_feats)) # shape (B, W, 255)
>
>         # Only retain the probability of the current stage
>         prob[:, stage_idx::n_stages] = stage_prob[:, stage_idx::n_stages]
>
>     return prob
> ```
>
> **2. Training Strategy**
>
> As discussed in Appendix B.2, we employ a mixed-stage training strategy by randomly sampling the number of stages S from a subset (e.g., {1, 2, 4, $W$}). Consequently, PACE generalizes effectively to unseen stage configurations at inference time. We attribute this to ECE’s node-level injection of sibling priors, which enables robust distribution estimation.
>
> **3. Optimization for Autoregressive Encoding**
>
> Currently, both the encoder and decoder use Mamba's single-step execution. An immediate optimization for encoding is adopting Mamba's convolutional parallel execution, akin to attention mechanisms in OctAttention and TopNet. Experiments show parallel execution reduces AR encoding to 1.73s but increases decoding to 34.61s, as the decoder must process the full window per node. As detailed in Appendix C, combining parallel encoding with serial decoding theoretically accelerates encoding without decoding overhead, but floating point discrepancies cause decoding failures. While model quantization [1] could resolve this, it requires substantial efforts, making it an independent research topic.
>
> [1] Yu J, et al. Mixed-precision post-training quantization for learned image compression. IEEE IoTJ, 2025.
>
> **4. Hyperparameters for GPE**
>
> The only hyperparameter in GPE is the number of neighbors $k$ used for the $k$-NN graph, which is set to 16 following prior work [2] [3]. New ablations varying $k$ show GPE is robust, achieving D1 BD-BRs of -50.24% ($k$=8), -50.19% ($k$=16), and -49.72% ($k$=32) in FP mode compared to G-PCC anchor.
>
> [2] Wang J, et al. A versatile point cloud compressor using universal multiscale conditional coding - part I: Geometry. IEEE TPAMI, 2025.
>
> [3] Wang Y, et al. Dynamic graph CNN for learning on point clouds. ACM TOG, 2019.
>
> **5. More Comprehensive Comparison**
>
> We thank the reviewer for the valuable comment and respectfully clarify:
>
> - For compression performance, Table 1 already provides comparisons with very recent methods published in 2025 and 2026, including RENO, NBP, TopNet, and LPCM.
>
> - We realize the concern may stem from our complexity analysis in Table 2, which currently focuses on open-source octree-based models. Here, we further expand this table to include other learning-based methods (see [anonymous link](https://anonymous.4open.science/r/icml_8078)).
>
> - We warmly welcome suggestions if we missed any specific concurrent works to include in the final version.
>
> **6. Extremely Sparse Scenarios**
>
> We conducted further experiments on the impact of point cloud sparsity. Due to space constraints, please refer to the section titled "Density-Controlled Experiments" in our response to Reviewer UxVL.
>
> We hope that this explanation addresses your concerns. Further implementation specifics (e.g., window partitioning details) will be included in the appendix. **The source code will be made publicly available.**

---

> > ### Author Rebuttal · Reviewer_Un7J · 2026-04-01
> >
> > We thank the author for their response and for the clarifications provided. After careful consideration, we have decided to maintain our original assessment and score. We appreciate the author’s effort in addressing our comments.

---

### Decision · Program_Chairs · 2026-04-30

**Decision:**

Accept (regular)

**Comment:**

The final ratings are 1 Accept (2TYR) and 3 Weak Accept (Un7J, KFTA, UxVL). The AC has read the reviews and the rebuttal.

The reviewers appreciated the motivation of the method (Un7J, KFTA, UxVL), technical design (Un7J, KFTA, 2TYR, UxVL), and thorough experiments (Un7J, KFTA, 2TYR). However, the reviewers also raised a number of concerns, including overclaim of suitability for practical application, but encoding time is 10.89 seconds (KFTA, 2TYR), incorrect use of non-causal terminology (KFTA), growth in latency and memory scaling with octree depth (UxVL), unclear limitation on sparse point clouds (UxVL), and lack of justification on different voxel precisions (UxVL).

The authors were able to address the comments from the reviewers and provided extensive experiments. Overall, the reviewers reached consensus for acceptance. The AC recommends the authors to consider the feedback and incorporate the materials presented in the rebuttal, which would improve the next revision of the manuscript.